# Estrogen-dependent control and cell-to-cell variability of transcriptional bursting

Christoph Fritzsch[1,†], Stephan Baumgärtner[1,†], Monika Kuban[1], Daria Steinshorn[1], George Reid[1,2,*,‡] & Stefan Legewie[1,**,‡]

## Abstract

Cellular decision-making and environmental adaptation are dependent upon a heterogeneous response of gene expression to external cues. Heterogeneity arises in transcription from random switching between transcriptionally active and inactive states, resulting in bursts of RNA synthesis. Furthermore, the cellular state influences the competency of transcription, thereby globally affecting gene expression in a cell-specific manner. We determined how external stimuli interplay with cellular state to modulate the kinetics of bursting. To this end, single-cell dynamics of nascent transcripts were monitored at the endogenous estrogen-responsive *GREB1* locus. Stochastic modeling of gene expression implicated a two-state promoter model in which the estrogen stimulus modulates the frequency of transcriptional bursting. The cellular state affects transcriptional dynamics by altering initiation and elongation kinetics and acts globally, as *GREB1* alleles in the same cell correlate in their transcriptional output. Our results suggest that cellular state strongly affects the first step of the central dogma of gene expression, to promote heterogeneity in the transcriptional output of isogenic cells.

**Keywords** bursting; estrogen signaling; heterogeneity; stochastic modeling; transcription

**Subject Categories** Quantitative Biology & Dynamical Systems; Transcription

**Mol Syst Biol. (2018) 14: e7678**

## Introduction

Heterogeneity is an essential feature of cellular decision-making. Genetically identical cells frequently respond in different ways to the same external stimulus, leading to differences in differentiation programs (Chang *et al*, 2008), drug resistance (Sharma *et al*, 2010; Paek *et al*, 2016), and viral pathogenesis (Weinberger *et al*, 2005). Such heterogeneous cellular behavior can be beneficial for the diversification of tissues and was shown to be related to variable expression of key regulators of cellular differentiation programs (Goolam *et al*, 2016).

Variability in protein levels arises because gene expression in single cells is a stochastic process (Harper *et al*, 2011; Suter *et al*, 2011; Molina *et al*, 2013). As a consequence of random, limiting, biochemical interactions, each gene has intrinsic temporal fluctuations in activity. For instance, mammalian transcription involves alternating transcriptionally active and inactive intervals, which are observed as transcriptional bursts (Chubb *et al*, 2006; Raj *et al*, 2006). Mathematical models have been developed that capture the stochastic nature of transcription and that interpret single-cell transcription datasets. In such models, promoters randomly switch between active (ON) and inactive (OFF) states (Paulsson, 2005; Suter *et al*, 2011; Zoller *et al*, 2015). The number of transcripts produced over time (i.e., the expression level) can be regulated by modulating burst frequency and burst size; that is, how often the promoter is active and by the number of transcripts produced per burst, respectively.

In addition to these intrinsic fluctuations, cells differ in their phenotypic state (e.g., cell cycle stage, cell volume, stimulation by extracellular conditions). This class of factors influences gene expression globally and introduces correlated fluctuations in multiple or in all genes. Such differences are referred to as extrinsic noise, and they can influence gene expression at various levels including transcription and translation. A unifying model that quantitatively describes gene regulation and that incorporates noise contributed by intrinsic and extrinsic factors is still lacking.

Live-cell microscopy of fluorescently labeled nascent transcripts provides a unique methodology to directly observe temporal fluctuations at the level of gene promoter activity. The PP7 reporter system enables such visualization and is based on the integration of PP7 sequences into a gene of interest. The PP7 sequences fold into stem-loop structures in nascent transcripts, which in turn associate with fluorescently labeled PP7 coat protein (PCP) (Chao *et al*, 2008). Observing transcription in four dimensions, that is, at distinct loci within multiple single cells over time, permits characterization of intrinsic and extrinsic variability.

1 Institute of Molecular Biology, Mainz, Germany
2 European Molecular Biology Laboratory, Heidelberg, Germany
  *Corresponding author. Tel: +49 6221 3878936; E-mail: george.reid@embl.de
  **Corresponding author. Tel: +49 6131 3921430; E-mail: s.legewie@imb-mainz.de
  †These authors contributed equally to this work as first authors
  ‡These authors contributed equally to this work as senior authors

In this work, we employed CRISPR/Cas9 genome engineering to introduce PP7 sequences into an endogenous *GREB1* locus. GREB1 is a central mediator of estrogen-induced cell growth *in vitro* and is a marker of tumor growth in estrogen-sensitive breast cancers (Rae *et al*, 2005; Laviolette *et al*, 2014). Estrogen (17β-estradiol, E2) activates transcription of target genes by binding to the ligand-dependent transcription factor estrogen receptor alpha (ERα). This signaling pathway is a paradigm for the dynamic behavior of chromatin in transcription (Métivier *et al*, 2003) and is a relevant mammalian system in which to study the adaptation of bursting to external cues.

We performed quantitative and time-resolved imaging of nascent *GREB1* transcripts to characterize how the dynamics of transcriptional bursting are modulated by E2 and by extrinsic noise sources. We employed a model fitting framework, known as approximate Bayesian computation (ABC), to calibrate stochastic models of transcription based on our data and to discriminate between alternative hypotheses of promoter regulation. We present a unifying model that quantitatively describes *GREB1* transcription as a two-state promoter cycle in which E2 regulates the frequency of transcriptional bursts. The cellular state modulates the amount of transcripts that are produced per burst by affecting kinetics of transcriptional initiation and elongation, thereby coordinately affecting multiple *GREB1* alleles in the same cell. Furthermore, we report that the relative importance of intrinsic and extrinsic noise sources can be altered by small-molecule inhibitors of histone deacetylases. In conclusion, our work quantifies how noise at different time scales is shaped by the contributions of transcriptional bursting, extrinsic noise, and the additive effects of multiple alleles.

# Results

### Direct observation of endogenous estrogen-mediated transcriptional activity

We wished to monitor endogenous estrogen-regulated transcription in living cells within a native chromatin environment. To achieve this, we modified a *GREB1* locus, using CRISPR/Cas9, in the ERα-positive breast cancer cell line MCF7 and visualized nascent transcripts using the PP7 reporter system. We generated the MCF7-GREB1-PP7 cell line by knocking-in an array of 24 PP7 sequences into exon 2, directly upstream of the start codon within a minimally perturbed *GREB1* gene (Fig 1A). Correct knock-in and recombination was confirmed by genomic PCR (Fig EV1A). Stable co-expression of the GFP-labeled PP7 coat protein (PCP-GFP) led to fluorescent labeling of nascent transcripts, with transcription sites visible as bright foci within the nucleus (Fig 1B and C). The presence of *GREB1* transcripts at these transcription sites was independently confirmed using single-molecule (sm) RNA fluorescence *in-situ* hybridization (FISH) with probes against intronic and exonic sequences of *GREB1* (Fig EV1D and E). The knock-in allele was transcribed at comparable levels to the two remaining endogenous *GREB1* alleles, as judged by exonic smRNA FISH spot intensities (Fig EV1G). Furthermore, the knock-in and wild-type alleles showed similar sensitivity to E2 stimulation in RT–qPCR analyses (Fig EV1B) and smRNA FISH (Fig EV1G). This suggests that the knock-in of PP7 sequences did not significantly perturb *GREB1* expression.

The mean intensity and frequency of occurrence of transcription sites increased in an E2 dose-dependent manner in the cell population (Figs 1D and EV1C and G). In the absence of E2, only a few cells had dim transcription sites, whereas at the saturating E2 concentrations, the *GREB1* locus was actively transcribed in around 90% of cells, highlighting an appropriate dynamic range within our experimental system. The pure anti-estrogen ICI 182,780 and the transcriptional inhibitor actinomycin D reduced spot intensities and the number of cells with active transcription, confirming that the occurrence of nuclear foci depends on estrogen signaling and on transcription (Fig 1D).

### Digital modulation of *GREB1* transcription by estrogen

Snapshot measurements at particular time points contain limited information about the kinetics of transcriptional bursting. We therefore monitored the temporal fluctuations of *GREB1* transcription for 13 h using time-lapse fluorescence microscopy at an imaging interval of 3 min, which is well below the estimated ~30 min residence time of individual, nascent *GREB1* RNAs at the locus (Fig 1E). Transcribing foci were detected, tracked within nuclei, and quantified from their 3D image volume (see Materials and Methods). Absolute transcript numbers were derived through calibration to the intensity of single transcripts from images at high excitation intensities (Fig EV1H). This quantification was independently confirmed through exonic smRNA FISH (Fig EV1F). We observed that endogenous *GREB1* is transcribed in stochastic bursts with up to ~150 elongating polymerases present on the body of the gene.

To evaluate the effect of E2 on burst properties, we recorded single-cell transcription after 3 days of stimulation at eight concentrations of E2, ranging from absence to saturating conditions (Figs 2A–C and EV2A–C, Dataset EV1). We analyzed about 60–90 cells per condition and observed that E2 increased the transcriptional activity of the *GREB1* gene in a dose-dependent manner. Dose dependence is also visible in the global intensity histogram over all cells and time points (Figs 2D and EV2D) as a characteristic bimodal distribution, in which transcription is either close to the background intensity or much higher, with intermediate intensities rarely observed. Similar bimodal distributions were also observed in smRNA FISH experiments (Fig EV2D). This suggests that *GREB1* exhibits digital ON/OFF-behavior, where increasing E2 increases the duration of time the gene spends in the transcriptionally active state. Higher doses of E2 furthermore gradually shift the right peak in the histogram toward higher intensity values. This suggests either an analog mode of transcription regulation, where more polymerases are recruited per burst, or an overlap in the signal between consecutive bursts after short OFF-times.

To further characterize the dynamics of E2-dependent regulation of transcriptional behavior, we directly extracted the duration of transcriptionally active and inactive periods from single-cell time courses by assuming that active periods are characterized by positive slopes in the time course (see Materials and Methods). We observed that the average pause duration in between bursts shortens with increasing E2 levels (from 184 to 26 min), while the average burst size increases (from 5 to 17 RNAs/burst) (Figs 2E and F, and EV2E and F). We thereafter employ mathematical modeling to quantitatively describe how burst properties change with the stimulus concentration.

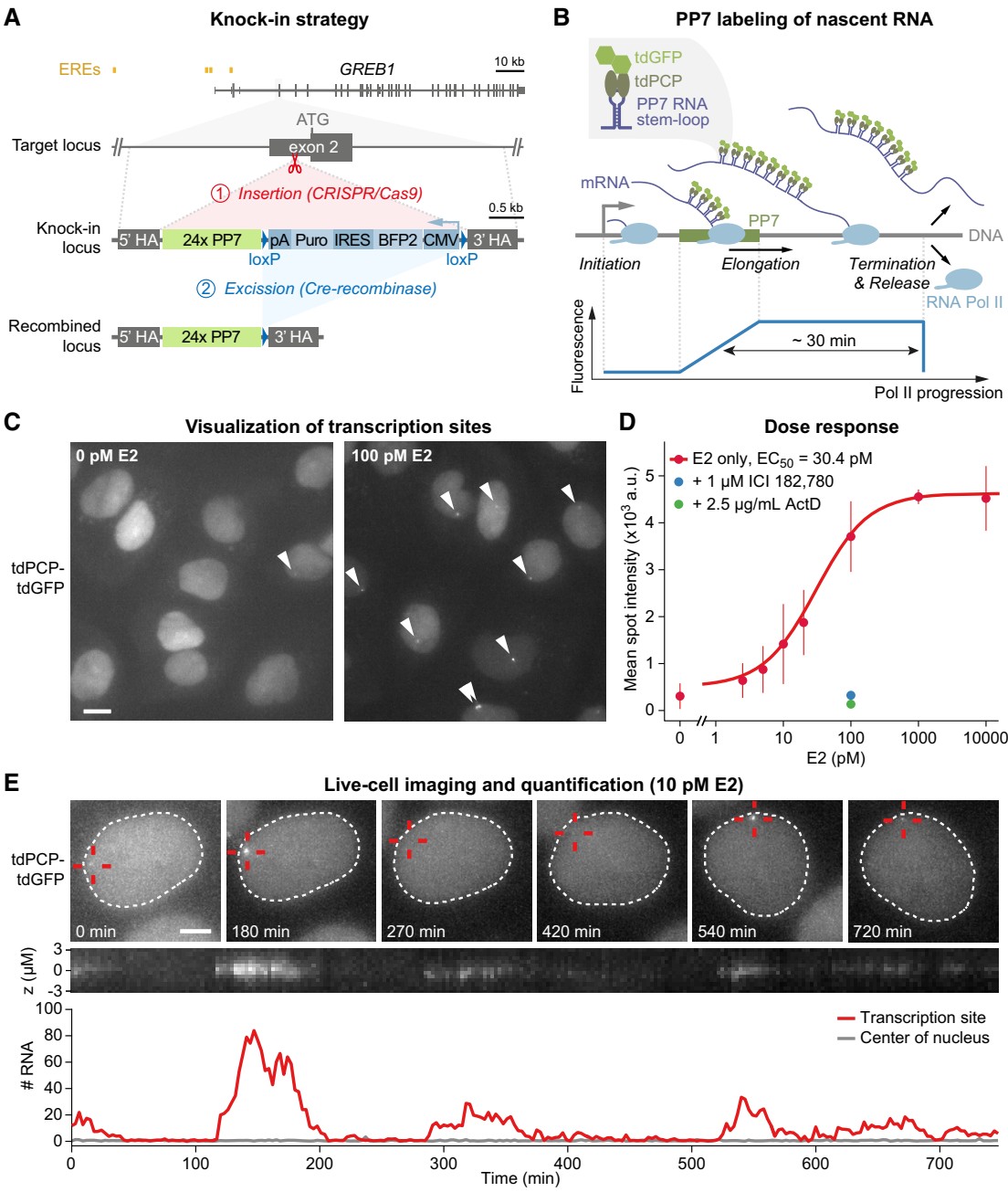

**Figure 1.  Knock-in of PP7 stem-loop sequences provides visualization of estrogen-mediated transcription from the endogenous *GREB1* locus in living cells (see also Fig EV1 and Movie EV1).**

A    Knock-in strategy to integrate PP7 sequences into a *GREB1* locus in MCF7 cells. CRISPR/Cas9-mediated knock-in of PP7 sequences, together with a selection cassette, into the 5′ UTR within exon 2 of *GREB1* was followed by excision of the selection cassette by Cre recombinase to yield the cell line MCF7-GREB1-PP7 (ERE: estrogen response element, HA: homology arm, pA: polyadenylation site, Puro: puromycin resistance, IRES: internal ribosomal entry site, CMV: promoter of cytomegalovirus).

B    Schematic description of the PP7 system. Binding of GFP-labeled PP7 coat protein (tdGFP-tdPCP) to PP7 stem-loops within nascent transcripts leads to fluorescence accumulation at the transcription site. Spot intensity decreases upon termination and transcript release. A schematic description of the fluorescence signal of a single transcript is shown below, with the 30 min which a transcript is observable estimated from gene length and published Pol II elongation rates.

C    Transcriptional foci in MCF7-GREB1-PP7 cells grown at low and high concentrations of E2. Single fluorescent foci (arrowheads) are observed within nuclei due to nuclear localization of tdGFP-tdPCP. Maximum intensity projections of *z*-stacks are shown. Scale bar: 10 μm.

D    E2 dose-response. Transcription sites were automatically identified and quantified in images of fixed MCF7-GREB1-PP7 cells. The mean ± standard deviation from three biological replicates of > 3,000 cells per condition is shown along with a fitted Hill function. ICI 182,780 (pure anti-estrogen) and actinomycin D (ActD; transcriptional inhibitor) serve to prevent transcription at 100 pM E2.

E    Endogenous E2-initiated transcription occurs in bursts. MCF7-GREB1-PP7 cells were imaged for 13 h at 10 pM E2. Transcription sites (red cross) were tracked within nuclei (dashed line). A zt-kymograph of the tracked transcription site demonstrates stable focus. Quantified transcript numbers are shown for a transcription site (red) and a control site at the center of the nucleus (gray). Scale bar: 5 μm.

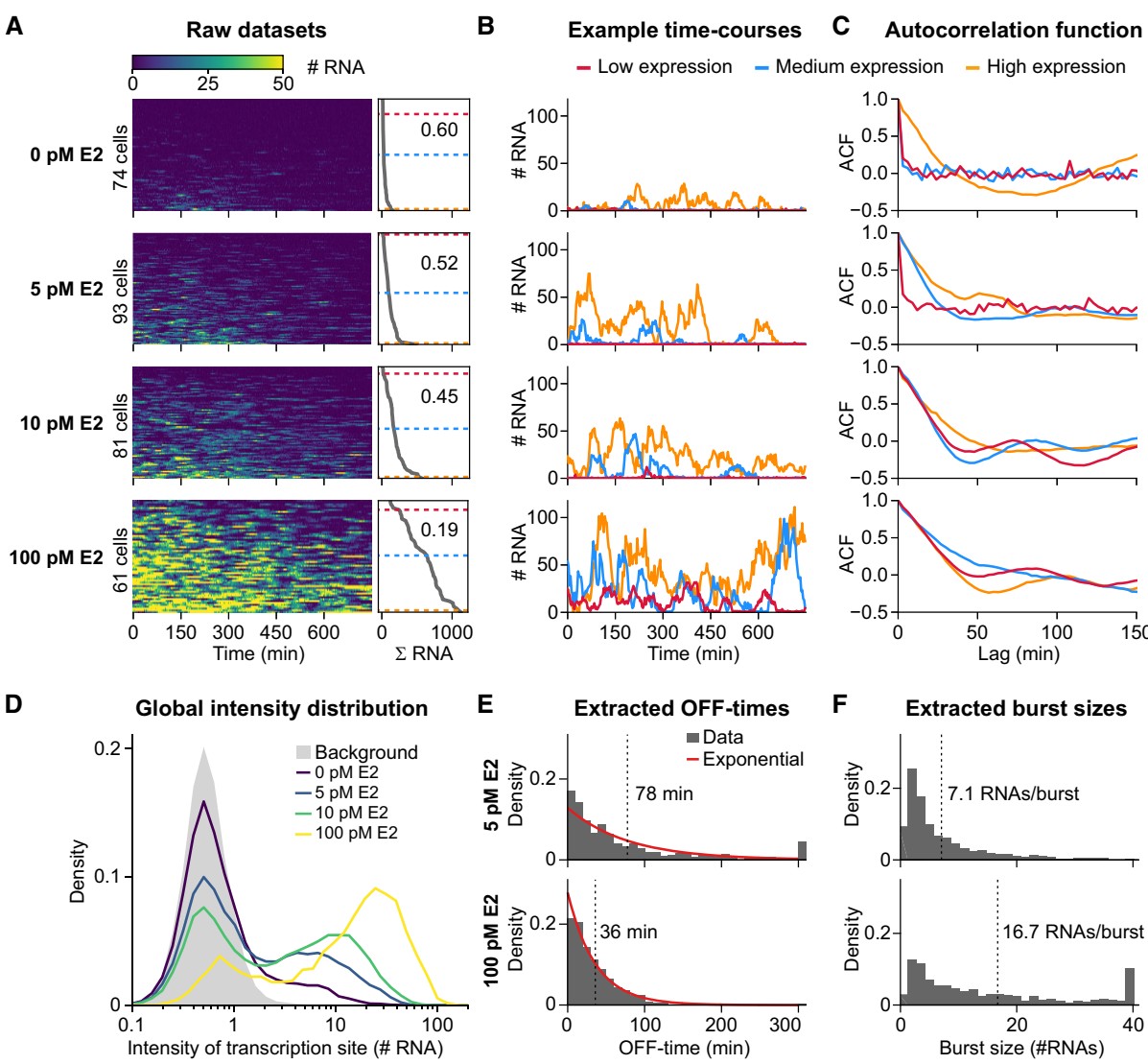

**Figure 2.  The transcriptional behavior of *GREB1* changes with estrogen dose and exhibits considerable cell-to-cell variation (see also Fig EV2 and Movie EV2).**

A    Cell-to-cell variation in *GREB1* expression at multiple E2 concentrations. Transcription was observed for 13 h in individual MCF7-GREB1-PP7 cells at different E2 concentrations. Trajectories of single cells are represented as color maps and are sorted from lowest (top) to highest (bottom) total RNA output (ΣRNA), calculated as area under the curve divided by the average signal from single transcripts (right). Color denotes the absolute number of nascent RNAs. The squared coefficient of variation (standard deviation²/mean²) of the total RNA output among the population is indicated (right). Dashed lines indicate exemplary cells shown in panels (B and C).

B    Representative time traces for low, medium, and high expressing cells.

C    Autocorrelation (ACF) curves of the individual time traces in (B).

D    Increasing E2 increases the proportion and productivity of transcriptionally active periods. Histograms were generated from the number of RNA molecules at the transcription site from all data points at different E2 concentrations. The distribution of the background signal is shown in gray.

E, F  Increasing E2 concentrations decrease the length of inactive periods and increase the burst size. Promoter OFF-times (E) and burst sizes (F) were extracted from the experimental tracks (see Fig EV2E). Exponential functions (red) with the same mean (dashed line) are shown for the OFF-times.

## The productivity of *GREB1* RNA synthesis exhibits considerable cell-to-cell variability

We observed that, although all individual cells show stochastic bursting, some cells generate low total amounts of RNA throughout the 13-h observation period, whereas others synthesize much larger total amounts of *GREB1* RNA (Fig 2A, right). This results in a considerable spread in the time-integrated intensity, which varies from 3,600 to 32,000 RNA·min between individual cells at a saturating E2 concentration of 100 pM. Considering the background intensity and that a single RNA contributes to the fluorescence signal for 30 min, this corresponds to a total RNA output of between ~110 and 1,100 *GREB1* RNAs within 13 h. The observed cell-to-cell variability is stable over time, as the RNA output during the first half of the movie correlates with the RNA output during the second half (Appendix Fig S1A). Thus, the *GREB1* locus

exhibits intrinsic stochastic dynamics and experiences more stable (extrinsic) fluctuations that affect long-term RNA production rates. This highlights that stable extrinsic factors, which reflect cellular state, directly impact gene expression on the level of nascent transcription.

Furthermore, at low E2 concentrations, we observe cells that do not have any transcriptional activation during the entire imaging period. Such cells can be reliably identified through analysis of their autocorrelation function, which decays instantaneously to zero if a trajectory consists solely of background noise (Figs 2C and EV2C). Based on this criterion, non-responders make up approximately 50% of the population in the absence of stimulation; all cells however respond when the concentration of E2 is above 10 pM. In responding cells, the autocorrelation function decays with slower kinetics ($t_{1/2}$ ~20–30 min), with a longer decay occurring with increasing E2 concentrations.

To investigate why individual cells show stable differences in transcriptional output, we extracted several morphological features from our microscopy images, namely cell area, nuclear shape, and local cell density. We found that none of these features alone correlated strongly and consistently across estrogen doses with overall transcriptional output (Appendix Fig S2). As a weak trend, we observe that cells with higher transcriptional activity tend to exhibit higher nuclear and cytoplasmic areas, supporting previous studies showing that the cell volume contributes to transcriptional output (Kempe *et al*, 2015; Padovan-Merhar *et al*, 2015). We consider it unlikely that the cell cycle stage constitutes a major source of extrinsic noise, as we completely discarded cells that show two transcription sites from a replicated allele at any time point during the observation period. Hence, cells that were analyzed never passed through S, G2 or M and were restricted to the G0/G1 phases of the cell cycle. Furthermore, we tracked cells after cell division and observed that stable differences in transcriptional output persist over the subsequent 6 h, when all cells are exclusively in early G1 phase (Appendix Fig S3 and Dataset EV2).

Taken together, our dataset exhibits several characteristic features, including a digital-to-analog global intensity histogram, strong cell-to-cell variability in the time-averaged intensity of transcription, and a subpopulation of non-responding cells. The interpretation of these phenomena is not straight-forward, due to the stochastic nature of the single-cell trajectories. We therefore turned to mathematical modeling to infer promoter dynamics from the data, and to understand why individual cells show a different susceptibility to estrogen-induced *GREB1* expression.

## Quantitative stochastic modeling of the dynamics of single-cell transcription

We implemented stochastic models that describe *GREB1* promoter activity and nascent RNA transcription. These models were fitted to experimental data to discriminate between different hypotheses of promoter regulation and to estimate kinetic parameters. As in previous work, the gene promoter was not modeled in molecular detail, but as an abstract cycle of transcriptionally active (ON) or inactive (OFF) promoter states (Paulsson, 2005; Zoller *et al*, 2015). As model variants, we considered five different promoter topologies, ranging from a simple two-state model with two rate-limiting

steps in gene (de)activation to a 10-state cycle (Fig 3A). A two-state model was sufficient to describe the behavior of other mammalian genes (Harper *et al*, 2011; Dar *et al*, 2012; Larson *et al*, 2013), while more states may better reflect the multiple sequential epigenetic steps reported for estrogen-dependent gene activation (Lemaire *et al*, 2006). Progression through promoter states was modeled to occur as a series of irreversible reactions with rates $k_{ON}$ and $k_{OFF}$, respectively, and transcription could be initiated from active states with rate $k_{init}$. We simulated the temporal evolution of the promoter using the stochastic simulation algorithm (Gillespie, 1977) and modeled polymerase-mediated transcript elongation deterministically with rate $k_{elong}$. To link model and experiment, we considered how each elongating transcript contributes to the fluorescent signal at the transcription site (Fig 3A, bottom).

The differences in the total *GREB1* RNA output observed between single cells (Fig 2A, right) could not be explained by the suggested models and consequently prompted us to add extrinsic noise sources into our model. We assumed extrinsic noise to be stable over time and implemented it by resampling selected model parameters before each single-cell simulation (Appendix Fig S1). For each of the five promoter topologies, we considered eight different extrinsic noise sources by resampling the elongation speed ($k_{elong}$), the transcription initiation rate ($k_{init}$), the promoter ON-/OFF-rates ($k_{ON}$, $k_{OFF}$), or combinations thereof. Each of those parameters represents a plausible target that could be influenced by the cellular state to alter transcriptional bursting.

For fitting the model to the experimental data, we used approximate Bayesian computation (ABC), as it provides a model fitting approach to simultaneously estimate model topologies and model parameters (Pritchard *et al*, 1999; Beaumont *et al*, 2002; Toni & Stumpf, 2009). We implemented a sequential Monte Carlo version of ABC (SMC ABC) which iteratively refines a population of 2,000 particles, each consisting of a combination of model topology and corresponding parameters, to yield measurement-compliant posterior distributions (Toni *et al*, 2009). The simulation result of each particle was compared to experimental data using a distance metric based on the global histogram of transcription site intensities, the autocorrelation function, and its heterogeneity to evaluate goodness of fit (Appendix Supplementary Methods). We benchmarked our SMC ABC framework using synthetic datasets with various combinations of promoter cycle topologies, extrinsic noise sources, and parameter values. We observed that kinetic parameter values and model structure are well recovered, though with some tendency to also retrieve closely related model topologies (Fig EV3A–C).

We fitted datasets from each E2 concentration separately and found that the best particles accurately recapitulate features of experimental data (Figs 3B and EV3D, and Appendix Fig S4) and yield posterior distributions of kinetic parameters (Fig EV3E). We analyzed the fitting results to determine which promoter cycle topology and extrinsic noise sources can describe the data (Fig 3C). Model selection strongly favored a two-state promoter cycle, consisting of a single rate-limiting step in promoter activation and deactivation. A two-state promoter model would predict that the OFF-times are exponentially distributed (Suter *et al*, 2011) and accordingly, we observed such a distribution in the OFF-times directly extracted from the data (Figs 2E and EV2F). As source of

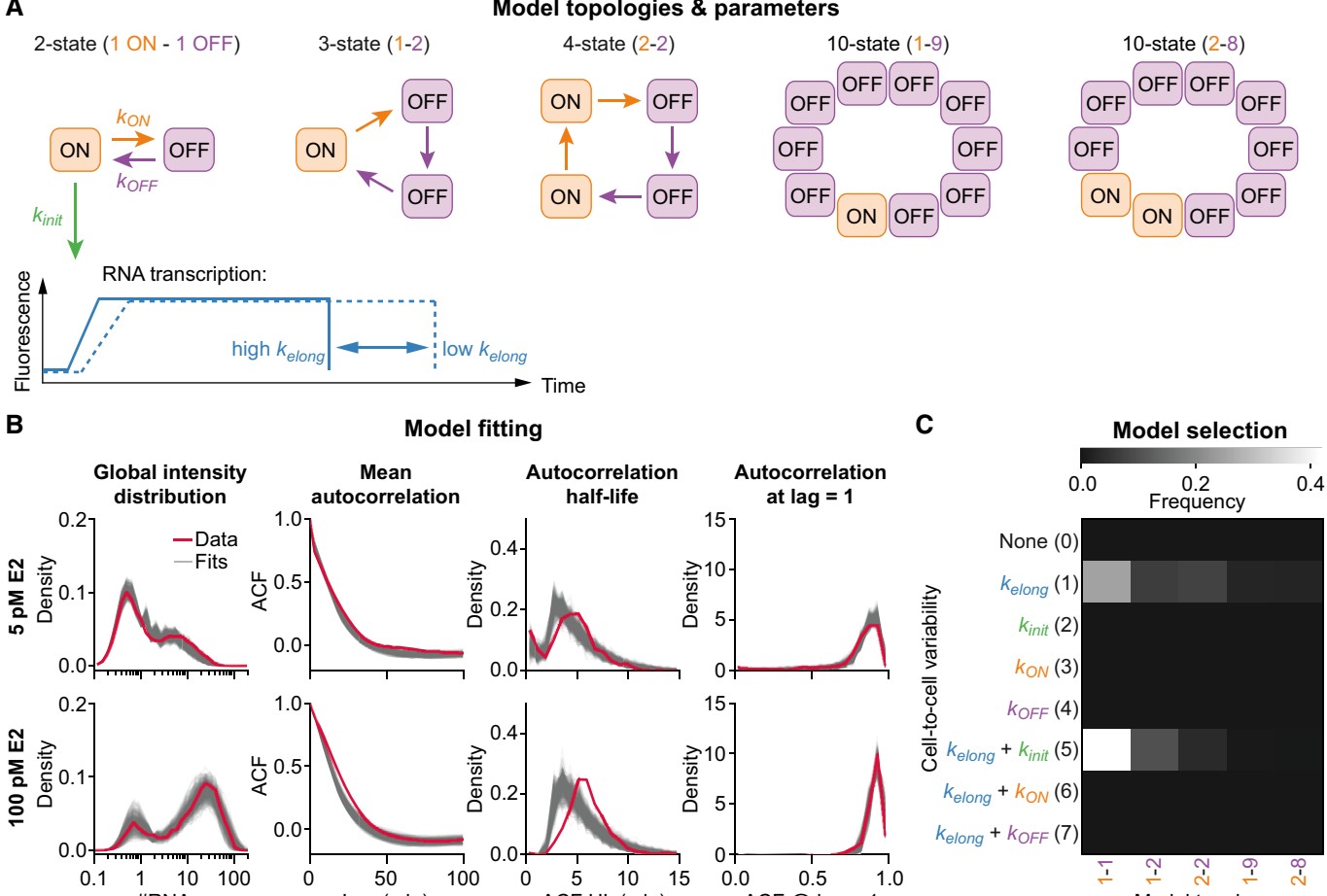

**Figure 3.  Stochastic model fitting can discriminate promoter complexity and cell-to-cell variability (see also Fig EV3).**

A   Topologies and kinetic parameters of models of different complexity. Top: The promoter cycles stochastically through active (ON) and inactive (OFF) states with rates $k_{ON}$ and $k_{OFF}$. New transcripts are initiated stochastically from the active state(s) with rate $k_{init}$. Extrinsic noise is implemented by cell-to-cell variability in model parameters (e.g., $k_{elong}$). Bottom: Signal of individual transcripts for fast (solid) and slow (dashed) elongation kinetics ($k_{elong}$).

B   Model fit to experimental data. Time course features used for model fitting are shown for datasets obtained at 5 pM (top) and 100 pM (bottom) E2 (red) and the 500 best particles obtained by model fitting (gray) (ACF: autocorrelation function)

C   Model selection favors a two-state promoter cycle with cell-to-cell variability in elongation and initiation rate. Posterior frequencies are shown for all 40 model topologies integrated over all eight E2 concentrations.

extrinsic noise, the model selection chose a combination of cell-to-cell variability of transcription elongation kinetics ($k_{elong}$) and initiation rate ($k_{init}$). This model variant, consisting of five parameters, was one of the few topologies observed in all fits (Fig EV3F) and made up as much as 42% of the fits when combining the particle populations of all E2 concentrations. By separately fitting two-state models with and without extrinsic noise sources to the data, we validated that extrinsic fluctuations in both the initiation and elongation rates are necessary to describe the experimental observations across a range of E2 concentrations (Fig EV3G and Appendix Fig S5). Hence, extrinsic noise sources, reflecting cellular state, alter two parameters of transcriptional bursts irrespective of stimulus conditions. This explains observed cell-to-cell variability and suggests that a unifying model can describe estrogen-mediated transcription. We next sought to validate the retrieved model structure by independent experiments.

**A simple promoter model is recapitulated in induction experiments**

The model prediction of a simple two-state promoter cycle suggests that the transitions between ON- and OFF-states occur with highly heterogeneous kinetics as opposed to extended promoter cycles with many states (Lemaire *et al*, 2006; Fig EV4A). Releasing cells from E2 starvation provides an experimental approach to determine the time it takes each individual cell to switch the gene from an OFF- to an ON-state and consequently, to estimate population heterogeneity in *GREB1* activation kinetics. Cells were starved of E2 for 3 days. After 51 min of imaging, transcription was induced by adding either 10 or 1,000 pM E2, following which *GREB1* nascent transcription trajectories were quantified (Fig 4A and Dataset EV3). Almost all cells showed transcriptional activation within 4 h, with a stronger response occurring at 1,000 pM E2, as expected based on steady-state

measurements (Fig EV2A). Furthermore, the response time for transcriptional activation was shorter at the higher E2 stimulus, in agreement with time-resolved RT–qPCR data (Fig EV4B). We observed highly variable response times for reactivation of transcription and no concerted initial transcriptional activation or subsequent coherent cycling between active and inactive states occurred. Thus, our single-cell data do not support a model in which estrogen target genes were postulated to be activated synchronously and rhythmically based on chromatin immunoprecipitation measurements (Métivier *et al*, 2003).

To determine whether the measured cell-to-cell variability under synchronization conditions can be quantitatively described by a two-state promoter model, we separately fitted small (two states, 1-1-5) and large (ten states, 1-9-5) promoter models to the synchronization data using SMC ABC. Synchronization was simulated by assuming that all cells are initially in the OFF-state and are simultaneously released upon activation of the *GREB1* gene. As expected, we found that the two-state model provided a better fit to the previously introduced data features than the ten-state model (Fig EV4C).

Simulations of the fitted two-state model are qualitatively similar to the experimental data, showing highly variable response times for transcriptional reactivation and no periodic transcription events (Fig 4B). To quantitatively compare model and data, we extracted response times from the time courses and calculated their coefficient of variation (CV = standard deviation/mean) as a measure of heterogeneity among the cell population. Both the experimental measurements and the two-state model had a CV close to unity (Fig 4C), as expected for a single rate-limiting step in gene reactivation in which response times follow an exponential distribution (Suter *et al*, 2011). In contrast, the ten-state model showed a substantially lower variability (CV = 0.6–0.7; see also Figs 4C and EV4D), consistent with a reduction of stochastic effects in sequential multistep regulatory processes (Lemaire *et al*, 2006). Taken together, these results suggest that only a few rate-limiting steps are needed to reactivate transcription from an OFF-state. Hence, the selection of small promoter cycles by the SMC ABC algorithm from steady-state measurements is plausible.

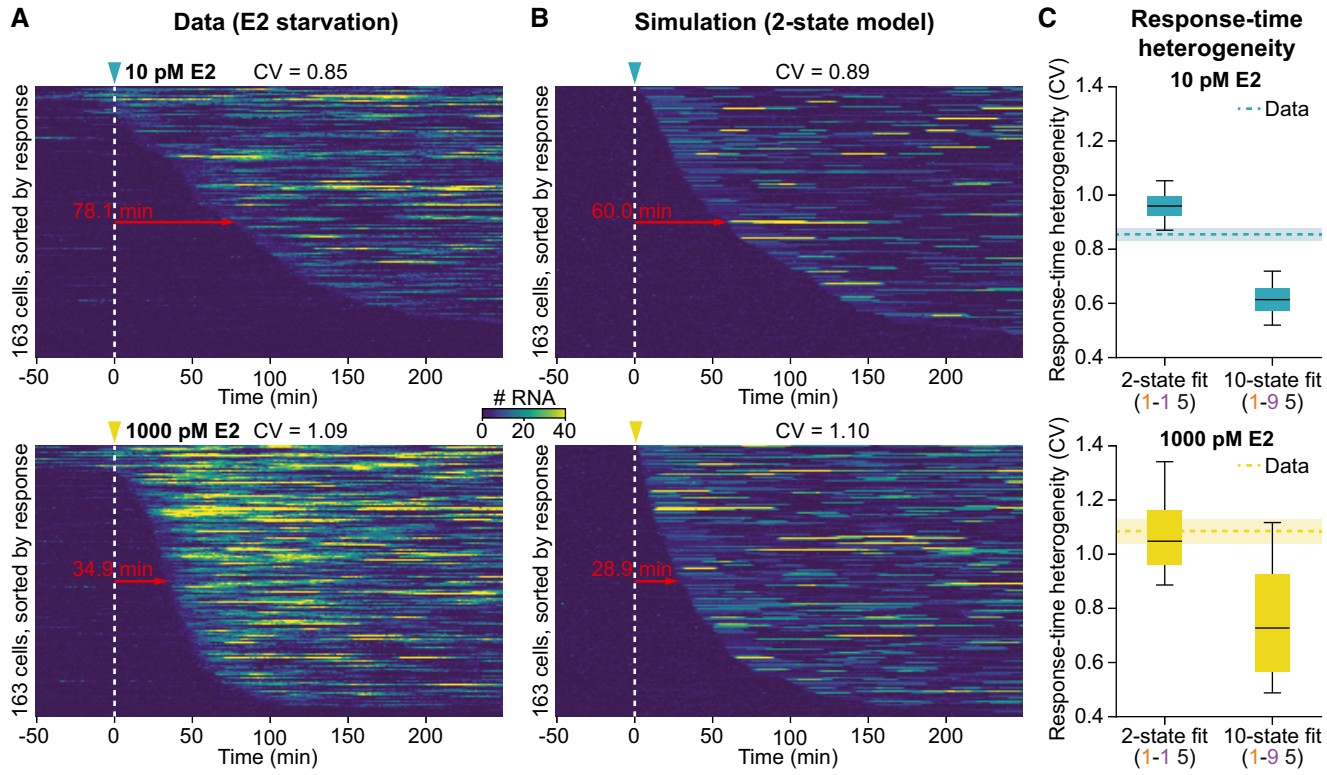

**Figure 4. Kinetics of *GREB1* induction by estrogen in single cells (see also Fig EV4 and Movie EV3).**

A   Response times after E2 induction are highly variable. Transcriptional trajectories of labeled *GREB1* loci upon addition of a sub-saturating (10 pM, top) and saturating (1,000 pM, bottom) E2 concentration were sorted based on their response time. MCF7-GREB1-PP7 cells were grown under E2-free conditions for 48 h prior to imaging. After 51 min of imaging, E2 was added at the indicated time point. Median response times are indicated (red arrow).

B   Simulated synchronization recapitulates variability in induction. Simulation of synchronized cells with parameters obtained from fits of the two-state (1–1–5) model to the 10 pM (top) and 1,000 pM E2 (bottom) synchronization datasets (panel A) produce trajectories that closely resemble the experiments ($t_{ON}$ = 0.8 min; $t_{OFF, 10\ pM}$ = 60 min; $t_{OFF, 1,000\ pM}$ = 30 min; $b$ = 6 RNAs/burst; model topology: 1–1–5).

C   Experimental response-time heterogeneity is explained by small promoter models. Response times were extracted from simulated induction experiments for all posterior particles of the SMC ABC fit (described in B). The boxplots show how the coefficients of variation (CV) in the response times (calculated over all cells of each simulated cell population) are distributed over the posterior particles (central line: median, box: 25 and 75% percentiles, whiskers: 5 and 95% percentiles). Experimental CVs are indicated as dashed lines with standard deviation from bootstrapping as shaded areas. A two-state model provides similar response-time heterogeneity as the data.

## Allele extrinsic sources dominate cell-to-cell variability

Our model suggests that the cell-to-cell variability in total *GREB1* RNA output can be explained by different rates of transcription initiation and elongation between cells. We next investigated whether these variations arise at the level of each individual allele (*cis*-acting, e.g., due to allele-specific chromatin states) or affect all alleles simultaneously (*trans*-acting, e.g., due to variability in cellular state). We established a further independent knock-in cell line that harbors two modified *GREB1* alleles, thereby permitting visualization of two estrogen-dependent loci within the same cell.

The MCF7-GREB1-PP7-Dual cell line was derived similarly to the single allele cell line, using CRISPR/Cas9-mediated knock-in and was validated by genotyping and smRNA FISH (Appendix Figs S6 and S7). To maintain protein function of the cell growth regulator *GREB1*, the PP7 sequences were delivered into intron 2 at two out of three *GREB1* alleles, about 1 kb downstream of the above-described knock-in site within exon 2. Transcription was recorded at 10 pM E2 (Fig 5A and C and Dataset EV4). Both alleles were visible as distinct fluorescent spots in the nucleus, with unrelated transcriptional bursting dynamics (Pearson correlation coefficient (PCC) = 0.01, Fig 5B). However, sister alleles were highly correlated in their RNA output (PCC = 0.54, Fig 5D), indicating a shared extrinsic noise source. No such correlation was observed for random pairs of alleles (PCC = 0.00). Furthermore, correlated total RNA output was observed between *GREB1* alleles of daughter cells within the first 6 h after division (PCC = 0.50, Appendix Fig S3 and Dataset EV2), indicating inheritability of factors that determine extrinsic noise. Taken together, these results suggest that stable extrinsic fluctuations in *GREB1* expression mostly arise from global cellular fluctuations of a *trans*-acting factor, while allele-specific variation is limited.

In order to compare these results to our fitted model, we performed dual-allele simulations in which the extrinsic noise source coordinately affects both sister alleles (Fig 5E). Specifically, sister alleles were simulated using the same values for the sampled transcription initiation rate and elongation kinetics. Under this assumption, both sister alleles showed independent stochastic bursting, but their total RNA output was correlated (PCC = 0.67, Fig 5F), in agreement with the experimentally observed correlation (Appendix Fig S8). No correlation was observed when the extrinsic noise source was eliminated from the model (PCC = −0.01). In conclusion, the predicted cellular fluctuations in transcription initiation and elongation rates are plausible.

## Estrogen modulates the frequency of transcriptional bursts

E2 increases *GREB1* transcription in a dose-dependent manner, with the posterior distributions of the individual fits (Fig EV3E) suggesting that it does so by increasing the frequency of transcriptional bursts, their size, or both. We refined our model fitting approach and asked whether we could simultaneously explain transcription at all E2 concentrations using a common model topology, assuming that E2 only modulates a single kinetic parameter.

Based on our previous fitting results, we fixed the model topology to a two-state promoter cycle with extrinsic variations in the transcription initiation and elongation rates. We found that the data at eight different E2 concentrations could be simultaneously fitted if we only allowed the promoter OFF-time to vary with the stimulus

(Fig 6A). This model outperformed a competing model variant in which E2 specifically modulates the burst size (i.e., the product of burst duration and number of RNAs produced per time unit in the ON-state). These fitting results hint at an estrogen-dependent modulation of burst frequency (Fig 6B) with *GREB1* being inactive for about 400 min in the absence of E2; this interval decreases to < 20 min at saturating E2 concentrations (Fig 6C). Given an average OFF-time of 400 min at low doses, a proportion of cells will not respond within an observation period of 750 min, in qualitative agreement with experiments (Fig 2A). When OFF-times are comparable to the residence time of fluorescent *GREB1* transcripts at the transcription site (~30 min), frequency modulation accounts for the analog shift of the right peak in the global intensity histogram, as consecutive fluorescent signals start to overlap (compare Figs 2D/ EV2D with Fig EV5D). For all levels of E2, the estimated ON-time is 0.56 min with 7.9 RNAs being produced per burst from the *GREB1* gene on average (Fig 6C).

Analytical calculations predict that transcriptional noise scales inversely with mean transcription for a frequency-modulated two-state model along lines of constant burst size (Elowitz *et al*, 2002; Singh *et al*, 2010). To compare these analytical results with our data, we separated the intrinsic and extrinsic noise components (see Materials and Methods) and observed that intrinsic noise follows this theoretically predicted noise–mean scaling (Fig 7A), thereby providing support for our global model.

In summary, quantitative modeling revealed that estrogen signaling acts upon the transition from an inactive to a transcriptionally permissive promoter state and has limited effect on polymerase loading during active periods. In such a regime of burst frequency regulation, noise is reduced with increasing estrogen levels.

## Burst properties and noise behavior are altered through protein acetylation

Previous studies reported that small-molecule inhibitors of epigenetic processes might regulate burst size and/or frequency (Suter *et al*, 2011; Vinuelas *et al*, 2013). Using live-cell imaging, we wished to directly characterize burst modulation by small-molecule inhibitors and asked whether noise in gene expression can be uncoupled from mean expression levels. Using high-content imaging of fixed cells, we observed that two inhibitors of zinc-dependent histone deacetylases (HDAC), the carboxylate butyrate and the hydroxamic acid trichostatin A, decreased total transcriptional output of the *GREB1* locus (Fig EV5A and B), as reported previously (Reid *et al*, 2005).

In order to quantify effects on transcriptional bursting, we performed live-cell imaging at intermediate concentrations of butyrate and E2 (Fig 7B and Dataset EV5). Interestingly, we observed that butyrate had no effect on gene expression noise at a low dose (2.5 mM), and only moderately increased noise at a higher dose (4 mM), while substantially reducing expression levels (Fig 7B–D, Appendix Fig S9 and Dataset EV6). Similarly, PFI1, a bromodomain inhibitor that blocks the recognition of acetylated moieties in chromatin, reduced *GREB1* expression, with moderate impact on noise (Appendix Fig S9 and Dataset EV6). Hence, these inhibitor effects did not follow the inverse noise–mean scaling exerted by E2 titration, implying that noise and mean can be controlled independently. Only intrinsic noise was affected by butyrate-mediated HDAC

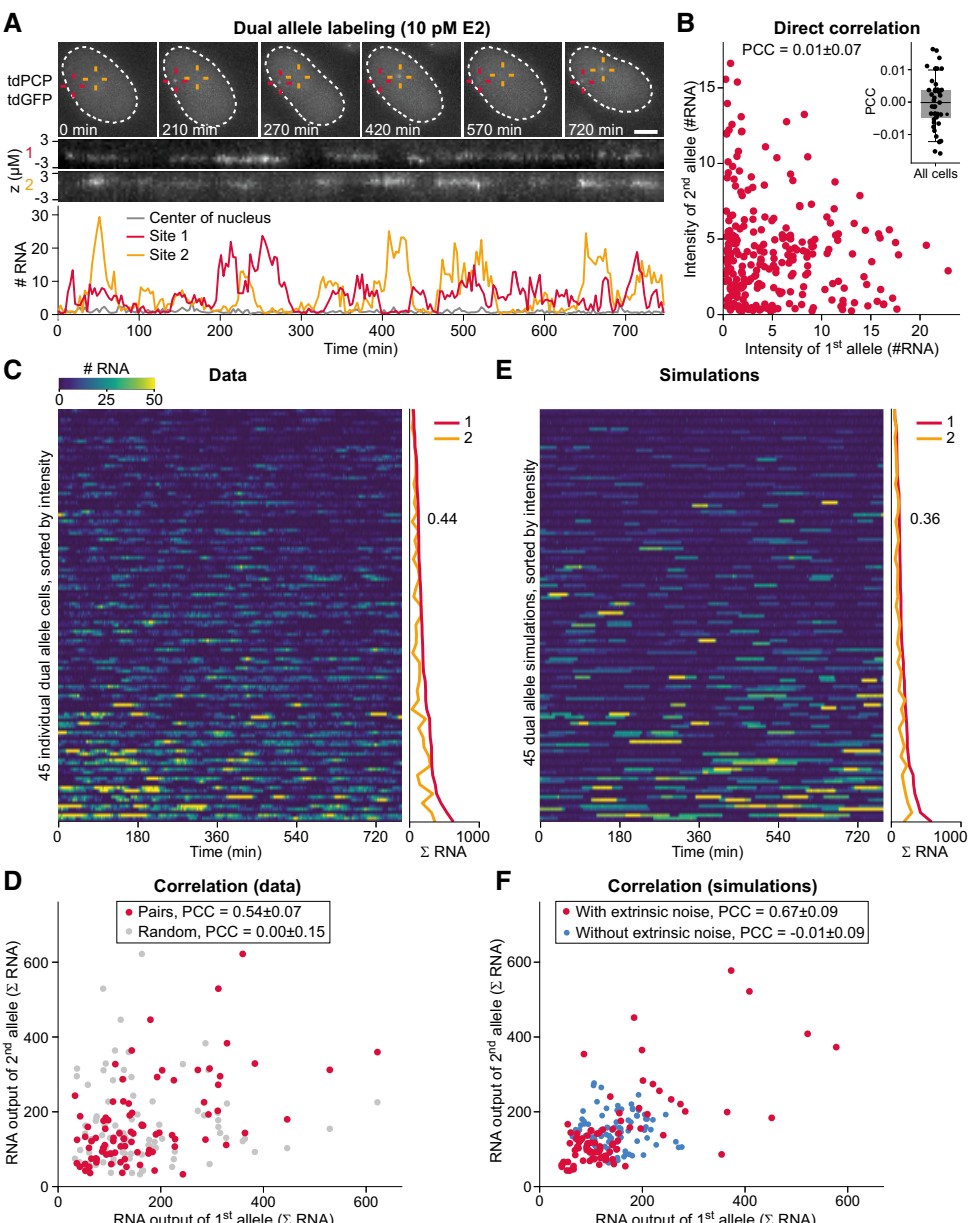

**Figure 5. Dual allele labeling reveals a *trans*-acting source for extrinsic noise.**

A   Observation of transcriptional bursts from two *GREB1* alleles in the same cell. MCF7-GREB1-PP7-Dual cells were grown at 10 pM E2. Two *GREB1* transcription sites (red and yellow crosses) were tracked in nuclei (dashed line), and the intensity of both transcription sites was quantified (bottom). The zt-kymograph demonstrates that both sites stay in focus during the movie. Scale bar: 5 μm. Also see Movie EV4.

B   Sister alleles have uncorrelated temporal fluctuations. Intensities of both transcription sites from the cell in panel (A) were plotted, with each dot representing one time point. The inset shows bootstrapped correlation coefficients for all 45 cells in panel (C).

C   Dual allele transcription in a cell population. Transcriptional activity was quantified in MCF7-GREB1-PP7-Dual cells at 10 pM E2. The signal of two alleles from the same cell is represented as pair of rows, separated by a dark line. Cells are sorted for the total RNA output (ΣRNA) of the brighter allele from low (top) to high (bottom) as indicated on the right. The squared coefficient of variation across all alleles is shown.

D   Total RNA output of sister alleles is highly correlated. The relationship of the total transcriptional output between sister alleles for all cells in panel (C) (pairs). Randomly reassigned alleles from different cells do not show correlation (random). To ensure unbiased calculation of the correlation coefficient, the dataset was doubled with each allele within one cell once assigned as allele one and once as allele two, causing symmetry around the diagonal axis.

E   Simulations recapitulate transcriptional profiles of dual allele reporter cells. Sister alleles were simulated using two stochastic simulations with the same extrinsic noise realization with parameters yielded by the model fit (Fig 3) to the steady-state 10 pM E2 dataset ($t_{ON}$ = 0.6 min; $t_{OFF}$ = 50 min; $b$ = 10 RNAs/burst; model topology: 1–1–5).

F   Simulations incorporating extrinsic noise recapitulate correlation in transcriptional output between alleles. Sister alleles were simulated using the same assumptions and parameters as in (E). The transcription initiation and elongation rates were (1–1–5 model topology; red) or were not (1–1–0 topology; blue) resampled for each cell to include/exclude extrinsic noise. The simulated alleles were treated the same as the real data to calculate correlation coefficients.

Data information: (D, F) Pearson correlation coefficients (PCC) are reported as mean ± standard deviation and were obtained via bootstrapping.

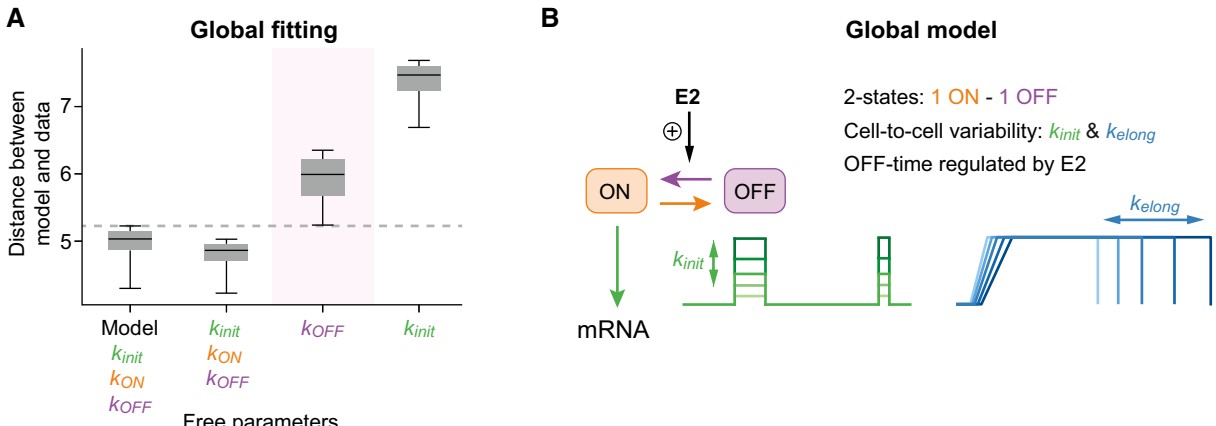

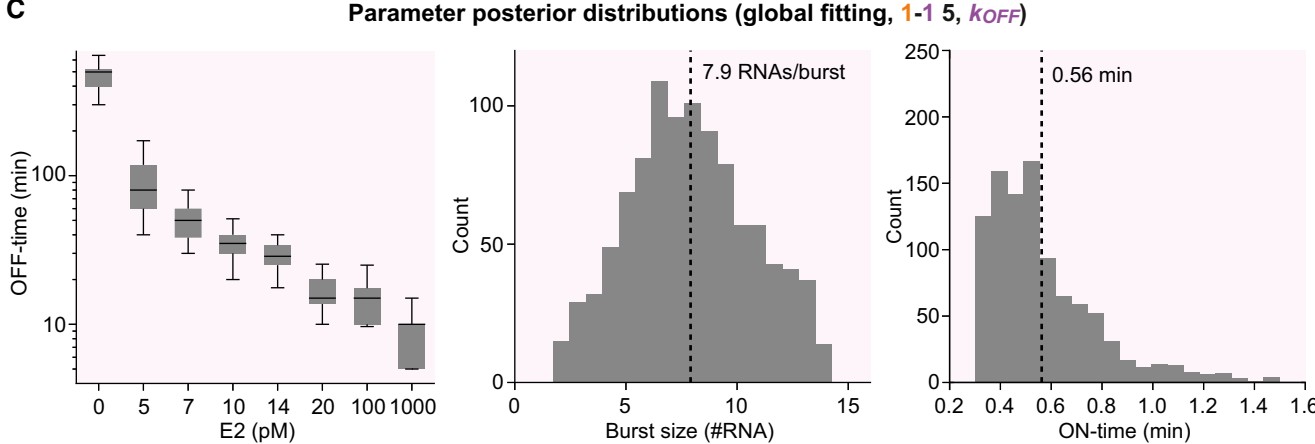

**Figure 6. Estrogen modulates the OFF-time between bursts.**

A　Global model fit to datasets at different E2 levels reveals stimulus-dependent frequency modulation. The distribution of summed particle distances when (1) fitting each E2 concentration with a separate parameter set (Fig 3) and variable model topologies (left); (2) fixing the model topology (1–1–5) and allowing all kinetic parameters to be different between E2 concentrations, (3) only allowing the OFF-time to change locally with E2 concentration and keep all other parameters global and (4) only allowing the transcription initiation rate to change locally (right).

B　Scheme of selected global model: Two-state model with burst frequency (i.e., OFF-time) modulation by E2 and cell-to-cell variability in initiation rate and elongation kinetics.

C　Parameter posterior distributions. Boxplots show posterior distributions of the promoter OFF-times obtained from the global SMC ABC fit with only the OFF-time being a local parameter (left). Histograms depict global posterior distributions and mean value (dashed line) for burst size (middle) and ON-time (right).

Data information: (A, C) Description of boxplots: central line, median; box, 25 and 75 percentile; whiskers, 5 and 95 percentile.

inhibition, while extrinsic noise was still comparable to that at an E2 concentration giving rise to the same output (Appendix Fig S10). According to analytical calculations, uncoupling of noise and mean can occur if a stimulus does not control the burst frequency, but instead regulates the burst size (Singh *et al*, 2010). To test whether our data support this hypothesis, we adopted the previously described fitting approach and separately calibrated our models using the DMSO and butyrate datasets. The posterior distributions confirmed that a low dose of NaBu (2.5 mM) primarily affects burst size when compared to DMSO control, while the higher dose additionally affects the burst frequency (Fig 7E). Burst features extracted directly from raw trajectories support this finding (Appendix Fig S11). To further support burst size modulation by butyrate, we performed a titration with this inhibitor using fixed-cell high-content imaging and extracted transcription site intensity histograms (Fig EV5C). At low doses, NaBu specifically affected the intensity of

transcription sites (reflecting burst size), but not their frequency in the cell population. These results were consistent with stochastic simulations of pure burst size modulation (Fig EV5D). Collectively, these data indicate that burst frequency and burst size can be orthogonally controlled to adjust intrinsic noise. Thus, protein acetylation levels allow for fine-tuning transcriptional output and associated noise independent of the estrogen stimulus.

## Discussion

### A two-state promoter model explains *GREB1* activation and deactivation

In this study, we quantified the dynamics of nascent transcription from an endogenous *GREB1* locus in individual breast cancer cells

and demonstrated that estrogen-controlled transcription occurs in bursts. Stochastic modeling, in combination with a Bayesian fitting approach, discriminated between a range of promoter model topologies with up to 10 states. Modeling predicted and experiments in synchronized cell populations supported the existence of two rate-limiting promoter states. A two-state model has also been used by others to describe transcriptional discontinuity (Peccoud & Ycart, 1995; Paulsson, 2005; Raj *et al*, 2006), although additional states were sometimes required to explain refractory periods in inactive promoter states (Harper *et al*, 2011; Suter *et al*, 2011; Zoller *et al*, 2015). It is surprising that a two-state model describes the estrogen response, a system that is known for multiple, ordered, cyclical and

sequential steps in gene activation (Métivier *et al*, 2003; Lemaire *et al*, 2006). We do not yet understand the discrepancy between such biochemical ensemble measurements of protein occupancy at the promoter and single-cell transcriptional activation. We speculate that only a subset of cells might contribute to the ordered effects observed at the population level, while single-cell studies observe the entire heterogeneous population.

### Estrogen modulates the frequency of transcriptional bursting

Gene regulation has been studied widely in the context of transcriptional bursting. Cells use a range of gene-specific burst sizes and

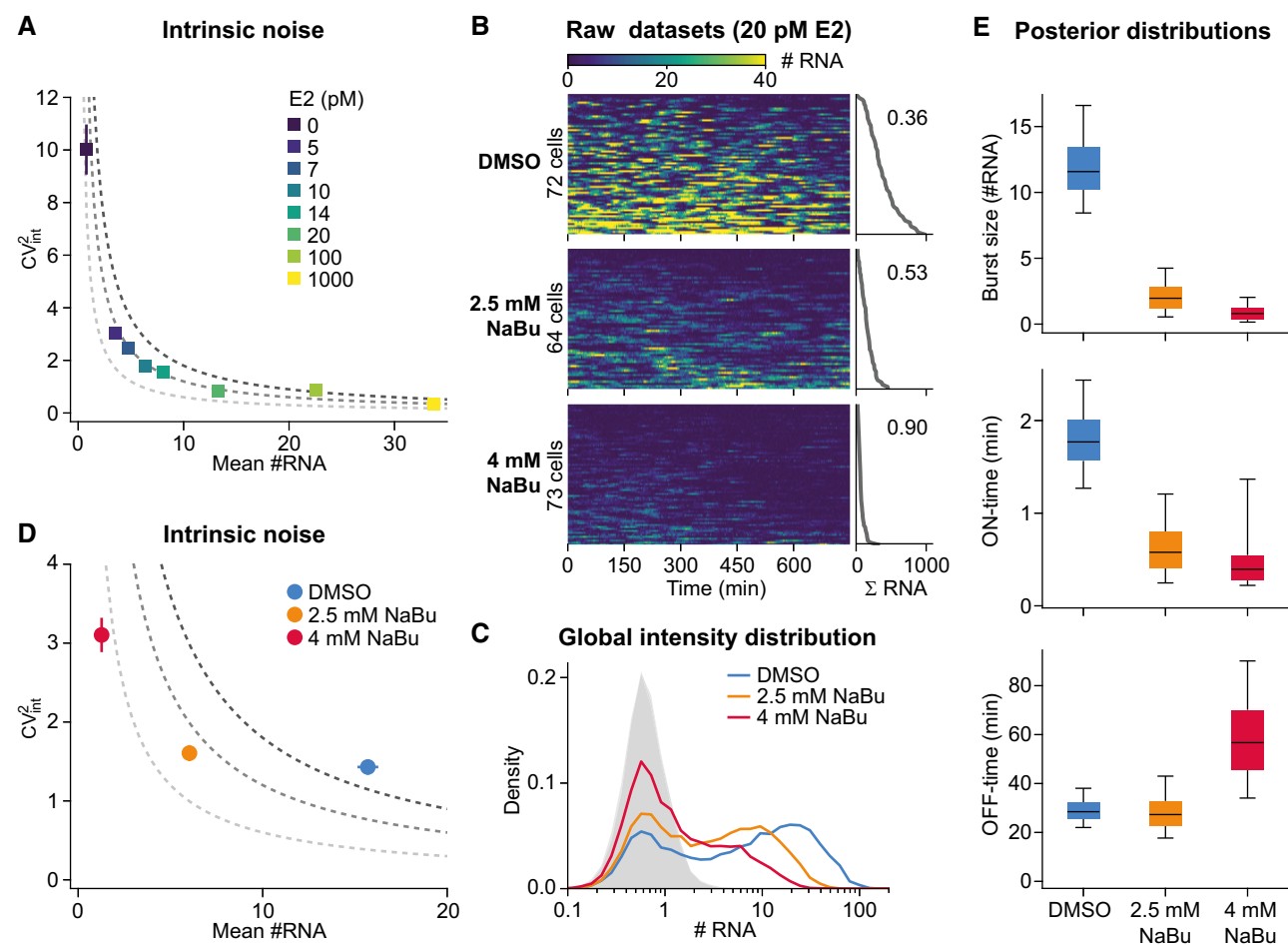

**Figure 7.  Modulation of protein acetylation perturbs estrogen-mediated transcription (see also Fig EV5).**

A   The dose-response to E2 shows an inverse noise–mean relationship. The measured intrinsic noise component (see Materials and Methods) is plotted as squared coefficient of variation against the mean expression level for each of eight datasets at various E2 concentrations. Dashed lines indicate the theoretical inverse noise–mean relation at fixed burst sizes, and changing burst frequencies ($CV^2$ = burst size/mean$_{RNA}$). Error bars represent standard deviation from bootstrapping.

B   Deacetylase inhibition reduces *GREB1* output. Transcription was quantified in cells exposed to 20 pM E2 after 4 h of pre-treatment with DMSO or butyrate (NaBu). Each line in the color maps represents an intensity trajectory from an individual cell. The total RNA output is plotted on the right with the $CV^2$ indicated.

C   Global intensity distributions for the datasets in panel (B). A shift to lower expression levels is apparent upon butyrate treatment.

D   Deacetylase inhibition affects noise in nascent *GREB1* transcription. The noise–mean relation is shown as in panel (A) for cells treated with 20 pM E2 or with solvent control (DMSO) or butyrate (NaBu). Butyrate leads to reduced *GREB1* expression at a lower noise level as compared to an E2 concentration with similar mean (E2 titration follows dashed lines, see panel A).

E   Butyrate predominantly reduces the burst size. Distributions of parameter posteriors from SMC ABC fits to DMSO and NaBu datasets (B) are shown as boxplots (central line: median, box: 25 and 75 percentiles, whiskers: 5 and 95 percentiles). A low dose of butyrate only affects burst size, while a higher dose also increases OFF-times.

frequencies to control expression levels (Suter *et al*, 2011; Dar *et al*, 2012). We observed that E2 increases *GREB1* transcription through reducing the duration of transcriptionally inactive phases. Stimulus-dependent frequency modulation was postulated decades ago (Moreau *et al*, 1981; Weintraub, 1988; Walters *et al*, 1995) and has recently been observed during MAPK induction (Senecal *et al*, 2014) and in early serum and TGF-β1 induction (Molina *et al*, 2013). However, these studies were limited to indirect observations using fluorescence *in situ* hybridization or labeling of proteins, which has only been recently complemented by direct observation of frequency-modulated nascent transcription upon stimulation of an artificial locus with insect hormones (Larson *et al*, 2013). By generating extensive, time-resolved datasets at multiple E2 concentrations, we provide definitive proof that frequency modulation occurs in an endogenous chromatin environment under native signaling. The frequency of bursts is dictated by the rate of formation of a transcriptionally competent initiation complex, suggesting that estrogen regulates the kinetics of gene activation, but not inactivation.

### Control of noise in transcription

In a frequency modulation regime, intrinsic noise in gene expression is coupled to its mean, such that lower noise is obtained with increasing expression levels (Singh *et al*, 2010). We could shift this noise–mean trajectory to lower noise levels through HDAC inhibition, which decreased burst size. This effect could be specific to a subset of genes, as intrinsic noise in *Nanog* expression for example is not affected by HDAC inhibition (Ochiai *et al*, 2014) and other genes show increased burst size (Harper *et al*, 2011). We anticipate that post-translational modification of chromatin delivers fine-tuning of transcriptional bursting and of consequential intrinsic noise. It is tempting to speculate that cells could utilize either frequency or burst size modulation in a gene-specific manner to achieve contrasting noise–mean scaling. A specific noise level could therefore be realized for each gene already on the transcriptional level and then be further modulated through downstream processes, such as RNA export, stability, and translation.

### Extrinsic noise shapes cellular heterogeneity

Our direct observation of nascent transcription enabled us to directly implicate transcriptional regulation as a major mediator of extrinsic variability, which propagates diversity in cellular states to temporally stable expression patterns. Other studies inferred intrinsic and extrinsic noise contributions by analyzing gene expression from two alleles at the mRNA (Raj *et al*, 2006; Gandhi *et al*, 2011) or protein (Elowitz *et al*, 2002; Raser & O'Shea, 2004; Harper *et al*, 2011) levels using snapshot measurements. Due to the relatively long half-lives and intermediate steps such as RNA processing and translation, time-averaging takes place which dampens intrinsic, bursting-related noise. Direct imaging of nascent transcripts at an endogenous locus requires long-term imaging and is inherently a low throughput procedure; however, it provides direct information on the origin of extrinsic noise without obfuscation from downstream processes. Stochastic modeling based on our data identified the kinetics of transcript initiation and elongation as major sources of variability, processes that have already been described to vary

between cells (das Neves *et al*, 2010; Annibale & Gratton, 2015; Sherman *et al*, 2015).

The question arises how initiation and elongation are regulated by the cellular state. Cell volume, metabolic state, upstream signaling, and microenvironment are candidate mechanisms that could globally influence initiation and elongation rates (Stewart-Ornstein *et al*, 2012; Battich *et al*, 2015; Padovan-Merhar *et al*, 2015). By extracting morphological features from our images, we could exclude that cell and nuclear size (both of which are presumably related to cell volume) are major determinants of extrinsic expression heterogeneity. This agrees well with a recent study which showed that the heterogeneity of housekeeping genes is strongly determined by cellular volume, whereas this is not the case for cell-type specific genes such as *GREB1* (Padovan-Merhar *et al*, 2015). Furthermore, we excluded the cell cycle stage as a relevant source of extrinsic fluctuations. It will be interesting to see in future immunofluorescence experiments whether the extrinsic noise in our data is related to stochastic fluctuations in the general transcription machinery, estrogen signaling components, or mitochondrial content. The differences in transcriptional output seem to be stable within our relatively long imaging time frame of 750 min. Therefore, we suspect that extrinsic differences will also propagate through to mRNA and protein levels, even when short-time fluctuations due to bursting are efficiently buffered by cellular systems (Stoeger *et al*, 2016).

Our study highlights how cellular state controls long-term transcriptional output, even in the presence of strong intrinsic fluctuations. Heterogeneous cellular states within a tissue are therefore a major determinant of expression variability. Consequently, the distribution in the level of growth regulators, such as *GREB1*, is broadened, with potential implications in heterogeneous growth phenotypes. Understanding determinants of cellular state within healthy and cancerous tissue and their effect on growth and variegated response to therapeutic intervention will be an important direction for future research.

## Materials and Methods

### Experimental methods

#### Cell culture

MCF7 cells were a gift from Edison T. Lui. All cell lines were maintained in Dulbecco's modified Eagle's medium (DMEM) with 4.5 g/l glucose (Lonza) supplemented with 10% fetal bovine serum (FBS-Gold, GE Healthcare), 1% L-glutamine (Lonza), and 1% penicillin/streptomycin (Lonza) at 37°C in a humidified atmosphere containing 5% $CO_2$. For experiments with controlled E2 concentrations, cells were grown in starvation medium composed of DMEM without phenol red (Thermo Fisher Scientific, Cat#31053-028), 2% charcoal-stripped FBS (Sigma-Aldrich, Cat#F6765), 1% L-glutamine (Lonza), 1% penicillin/streptomycin (Lonza), and the defined amount of 17β-estradiol (Sigma-Aldrich, Cat#E8875).

#### Generation of reporter cell lines

To achieve stable expression of the GFP-labeled PP7 coat protein, MCF7 cells were transfected with 3 μg of pSB-Ubc-NLS-HA-tdPCP-tdGFP and 1.5 μg of pCMV(CAT)T7-SB100X (gift from Zsuzsanna

Izsvak, Addgene plasmid #34879) using FuGENE® HD Transfection Reagent (Promega, Cat#E2311). Single cells with low GFP signal were isolated after 14 days by flow cytometry on an Aria III SORP (Becton Dickinson) giving rise to the clonal cell line MCF7-SBtdPCPtdGFP. The same procedure was carried out on MCF7-GREB1-PP7-Dual_noPCP cells to yield MCF7-GREB1-PP7-Dual.

MCF7-SBtdPCPtdGFP cells were transfected with 1.5 μg of pX330-GREB1-ex2 and 3 μg of pHR-GREB1-ex2-24xPP7-LPIBCL with FuGENE® HD and selected with 0.1 μg/ml puromycin (Thermo Fisher Scientific, Cat#A1113803). Clonal colonies were picked using cloning cylinders, screened for the presence of eBFP2 labeled peroxisomes, and genotyped using genomic PCRs, giving rise to the MCF7-GREB1-PP7-BFP-Puro cell line. A similar knock-in strategy was performed for intron 2 of *GREB1*, by transfecting pX330-GREB1-int2 and pHR-GREB1-int2-24xPP7-LPIBCL into MCF7 cells to yield MCF7-GREB1-PP7-Dual_noPCP.

MCF7-GREB1-PP7-BFP-Puro cells were transfected with 3 μg pCAG-Cre-IRES2-GFP (gift from Anjen Chenn, Addgene plasmid #26646) and sorted for BFP/GFP double-positive cells. Single BFP-negative cells were isolated by flow cytometry after 3 weeks. Excision of the selection cassette in the resulting MCF7-GREB1-PP7 cells was confirmed by genotyping PCRs.

### Quantification of gene expression by RT–qPCR

$2 \times 10^5$ cells were seeded into a 6-well plate and harvested after 3 days of growing at different conditions. Transcription was induced overnight (~18 h) for the E2 dose-response curve. Alternatively, cells were induced with either 10 or 1,000 pM E2 and samples were collected every 10 min for 2 h to measure kinetics of RNA induction. 600 μl of TRIzol® reagent (Thermo Fisher Scientific) was used for RNA isolation according to manufacturer's instructions. Reverse transcription was carried out on 400 ng of total RNA in a volume of 12 μl using the SuperScript® II RT (Thermo Fisher Scientific, Cat#18064014) with random hexamers according to manufacturer's instructions. qPCRs were performed on 1 μl of cDNA with 100 nM of each primer within a 10 μl reaction using Power SYBR Green PCR Master Mix (Thermo Fisher Scientific, Cat#4364344). All measurements were performed as technical duplicates in a ViiA™ 7 Real-Time PCR System (Thermo Fisher Scientific) in 384-well plates (Roche) using 40 cycles of a two-step protocol with 15 s at 95°C and 1 min at 60°C. Efficiency of amplification was determined from serial dilutions of template DNA and confirmed to be above 90% for all primer pairs.

Mean Ct-values of technical replicates were used. Gene expression was quantified after normalizing to the reference gene *GAPDH* using the ΔΔCt method.

During allele-specific RT–qPCR for wild-type and knock-in allele, normalization was performed through a serial dilution of pUC-qRT-GAPDH-GREB1ex2-wt-PP7 plasmid, which contained all PCR products.

The RT–qPCR E2 dose-response was performed in quadruplicates and fitted to a four-parameter Hill equation using the "nls" function in R. E2 induction was carried out in triplicates.

### High-throughput imaging

$1 \times 10^4$ cells were seeded per well of a 96-well SensoPlate™ Plus glass bottom microplate (Greiner, Cat#655891) 3 days prior to image

acquisition. The medium was replaced with medium containing the desired concentration of E2 every day, and DMSO (Sigma-Aldrich) or small-molecule inhibitors (actinomycin D, Sigma-Aldrich, Cat#A1410; sodium butyrate, Sigma-Aldrich, Cat#303410; trichostatin A, Sigma-Aldrich, Cat#T1952; ICI 182,780, Selleckchem, Cat#S1191) were added 4 h prior to fixation. Cells were washed with PBS, fixed with 4% PFA in PBS for 10 min on ice, and washed twice with PBS, and a nuclear counter-staining was performed with 0.5 μg/ml DAPI (Sigma-Aldrich) in PBS for 5 min, or 2.5 μM DRAQ5™ (eBioscience) for 30 min. Cells were stored in PBS at 4°C until image acquisition.

Seven fields were imaged per well in an Opera Phenix™ High Content Screening System (PerkinElmer) with a 20× 1.0 NA water immersion objective using spinning disk confocal mode as a z-stack with 22 planes spaced 1.2 μm apart without binning, resulting in a pixel size of 0.30 μm. Exposure time in the eGFP channel (excitation: 488 nm laser, emission: 500–550 nm) was 500 ms at 100% illumination intensity. Nuclei were imaged either in the DAPI channel (excitation: 405 nm laser, emission: 435–480 nm) for 60 ms at 80% intensity or in the DRAQ5 channel (excitation: 640 nm laser, emission: 650–760 nm) for 300 ms at 50% intensity, depending on the nuclear counterstain. Images from the high-content screening microscope were analyzed using the Harmony® High Content Imaging and Analysis Software (PerkinElmer) with a custom pipeline (see Appendix Supplementary Methods for details).

High-content imaging for the E2 dose-response was performed in three separate experiments with technical duplicates. Inhibitor treatment was performed twice with technical duplicates.

### Live-cell imaging

$1.8 \times 10^4$ cells were seeded into a channel of a μ-Slide VI 0.4 ibiTreat (Ibidi) slide 3 days prior to imaging. Medium was replaced daily with starvation medium containing the desired amount of E2 until imaging. For inhibitor treatments, 0.05% DMSO (Sigma-Aldrich) with or without sodium butyrate (Sigma-Aldrich, Cat#SML0352) or PFI1 (Sigma-Aldrich, Cat#SML0352) was added 4 h prior to imaging after growing cells in 20 pM E2 for 2 days. For induction experiments, cells were grown in starvation medium without E2 for 48 h and placed into the microscope. After 51 min of imaging, the medium was replaced with starvation medium containing either 10 or 1,000 pM of E2.

Live-cell images were acquired on a DeltaVision™ Elite microscope system (GE Healthcare Life Sciences) equipped with an environmental control chamber (Imsol) and a $CO_2$ mixer (Leica) to maintain 37°C and 5% $CO_2$ during imaging experiments. Excitation light was generated using a seven-color InsightSSI module and focused through an 60× 1.42 NA Oil Plan APO objective. Excitation and collection of eGFP fluorescence was achieved using the FITC filters and the polychroic beam splitter for DAPI, FITC, TRITC, and Cy5. Images were acquired on a pco.edge sCMOS-camera operating in $2 \times 2$ binning mode, yielding a pixel size of 216 nm. The microscope was controlled via softWoRx software in version 6.5.2.

Z-stacks with 12 planes spaced 0.55 μm apart were acquired every 3 min for 260 time points (total imaging time ~13 h) in the FITC channel with 2% light intensity with 100- or 120-ms exposure time. One brightfield image (50-ms exposure, 5% light intensity) was acquired at each time point to follow cell viability. The first 10

frames of each movie were discarded to avoid initial photobleaching of the medium. For induction experiments, images were acquired with the same settings every 1.5 min for 200 time points (300 × 5 min for analysis of daughter cells) without discarding initial images. Images for visualization of single transcripts were acquired as a z-stack with 14 slices spaced 0.27 μm apart, with 100% light intensity and 100- or 120-ms exposure time.

Imaging of the E2 dose-response was performed for the 0–20 pM E2 datasets simultaneously on the same day and for 100–1,000 pM E2 datasets on a different day. Data for E2 induction are a combination of two separate experiments for each E2 concentration. The dual allele dataset, the daughter cell dataset, and the inhibitor dataset are from one experiment each.

### Single-molecule RNA FISH

Stellaris® FISH probes recognizing exons 5–9 of human *GREB1* were obtained from Biosearch Technologies labeled with Quasar® 570 (#VSMF-2158-5). Custom Stellaris® FISH Probes were designed against introns 2, 9, and 10 of human *GREB1* by utilizing the Stellaris® RNA FISH Probe Designer (version 4.2) (Appendix Table S4) and labeled with Quasar® 670. Both probe sets were hybridized to cells following the manufacturer's instructions for adherent cells with adjusted volumes of reagents to adapt to the use of channel slides. Briefly, $1.8 \times 10^4$ cells were seeded into μ-Slide VI 0.4 ibiTreat slides (ibidi), grown with different E2 concentrations for 3 days, and fixed with 4% PFA for 10 min. Cells were washed twice with PBS and permeabilized with 70% ethanol at 4°C for at least 1 h. Cells were rehydrated in wash buffer A and hybridized in 50 μl hybridization buffer with 125 nM of each probe at 37°C overnight. Cells were washed twice with wash buffer A for 30 min at 37°C (second wash with 5 ng/μl DAPI), once with wash buffer B, mounted in oxygen-depleted medium (according to Raj et al, 2008), and imaged immediately.

Images were acquired on a DeltaVision™ Elite microscope system described above. Acquisition was performed without binning, yielding an image with 2,048 × 2,048 pixels and a pixel size of 108 nm. z-stacks with 32 planes spaced 0.27 μm apart were acquired in the TRITC channel (50% light intensity, 200-ms exposure) for Quasar® 570, the Cy-5 channel (32% light intensity, 100-ms exposure) for Quasar® 670, the FITC channel (100% light intensity, 100-ms exposure) for GFP, and the DAPI channel (50% light intensity, 100-ms exposure). Image analysis was performed with custom MATLAB scripts (see Appendix Supplementary Methods for details).

### Live-cell image analysis

All live-cell image analyses were performed using custom MATLAB scripts (see Appendix Supplementary Methods for details). Briefly, nuclei were segmented from maximum intensity projections and tracked throughout the movie. Nuclear spots were identified on maximum intensity projections of bandpass-filtered images and tracked using the u-track package (Jaqaman et al, 2008). Resulting short tracks were linked to generate full-length tracks, and erroneously assigned positions were corrected manually. Transcription sites were relatively immobile with respect to the nucleoplasm, allowing for successful tracking relative to the nuclear movement, even in the absence of visible spots. If no spot was detectable throughout the whole movie, a position close to the center was used. Cells in which the transcription site moved out of focus or

duplicated during acquisition were discarded. Spot intensities were quantified by fitting a three-dimensional Gaussian distribution.

### Absolute quantification by calibration to intensities of single RNAs

MCF7-GREB1-PP7 cells grown at 1,000 pM E2 were imaged at 100% excitation light. Dim spots that likely represent single transcripts were manually identified and quantified as described above. A Gaussian distribution was fitted to the resulting intensity histogram, and its mean was used as the intensity for a single transcript. To adjust for the relative difference in illumination between 2 and 100% light intensity, images were acquired with both illumination conditions at the same position, intensities of transcription sites were quantified, and a linear function was fitted to the ratio of intensities to yield a normalization factor. The procedure was repeated every time the setup of the microscope or image acquisition parameters changed.

### Feature extraction from time traces

A running median with a window size of five time points was applied to smooth the time trace. The slope of the curve was calculated as the difference between two consecutive time points and subsampled 10-fold by linear interpolation. A threshold of 0.65 transcripts per 3 min was applied. Gaps and peaks with a duration of less than one imaging interval were discarded. ON- and OFF-times were derived from the time the slope is above or below the threshold, respectively. The burst size was calculated for each ON-period as the difference of intensity of the smoothed time trace. The initiation rate was calculated by dividing the burst size by the ON-time. ON- and OFF-times were also calculated for regions that encompass beginning or end of the time trace, to include long OFF-times for non-responders at low E2 concentrations.

Response times were calculated from induction experiments and simulations by median filtering with a window size of seven time points and determining the time from which the trajectory stayed above a threshold of two transcripts for at least five consecutive time points.

## Computational methods

### Model parametrization and stochastic simulation

We implemented hybrid stochastic-deterministic simulations of the models shown in Fig 3A. All model topologies share the main parameters promoter ON-time $t_{ON}$, OFF-time $t_{OFF}$, and burst size $b = k_{init}t_{ON}$, with $k_{init}$ being the transcription initiation rate from an active promoter. For longer promoter cycles with multiple ON- or OFF-states, the total time is split between those states (see Appendix Supplementary Methods for details). Switching between promoter states and time points of transcriptional initiation is simulated using the stochastic simulation algorithm (Gillespie, 1977). Transcript elongation is modeled deterministically, and the signal of a single transcript has the profile depicted in Fig 1B: Due to the position of the PP7 stem-loops in the gene, the fluorescent signal of a single transcript gradually appears after a first delay following transcript initiation and disappears after a second delay when transcript termination occurs. The output of the simulation is the sum of fluorescence intensities of all elongating transcripts at a certain time point. Simulations thus yielded time courses of absolute RNA numbers at the transcription site and were connected to

experimental data using a noise model (see Appendix Supplementary Methods).

As cells exhibited substantial cell-to-cell variability, we included various potential sources of extrinsic noise into the models, each of which influences specific kinetic parameters of the model. We implemented this numerically by resampling the actual parameter for each single cell from a normal distribution around the population mean parameter value. The width of this distribution is an extra parameter describing the strength of the cell-to-cell variability. In dual-allele simulations (Fig 5E), we used the same parameter realization for both alleles of the same cell.

### Approximate Bayesian computation to calibrate stochastic models

Approximate Bayesian computation (ABC) was used to estimate model parameters and topologies based on the data (Tavaré *et al*, 1997; Beaumont *et al*, 2002). We used five different features to compare data and simulations, including (i) the global intensity histogram, (ii) the mean autocorrelation over all cells in the population, (iii)-(iv) the single-cell distributions of the autocorrelation half-lives and autocorrelations at lag 1, respectively, and (v) the maximum-mean discrepancy. See Appendix Supplementary Methods for a description of how these features were converted into a quantitative distance metric between model and data.

We followed a Sequential Monte Carlo Approach of ABC in which a population of 2,000 particles, each representing a model and the corresponding parameters, are refined iteratively. In an initial iteration, candidate particles are sampled from prior distributions and simulations of each particle are compared to experimental data using the distance metric. In subsequent iterations, new particles are generated by randomly sampling the parameter space in the vicinity of the 20% best existing particles using proposal distributions for each parameter individually. Proposal distributions depend on the current particle position and a predefined width which was fine-tuned in algorithm benchmarking. The new particle population is assembled from at least the 20% of the best particles of the previous iteration and new particles that provided an improved description of the data. The algorithm stops when there is no significant improvement between consecutive iterations. Benchmarking using synthetic datasets confirmed that the algorithm can distinguish model topologies and correctly recovers parameters (see Appendix Supplementary Methods, and Fig EV3A–C). The same fitting approach and distance metric were used to fit steady-state *GREB1* expression at different E2 concentrations (Fig 3), steady-state *GREB1* expression upon inhibitor treatment (Fig 7), and induced *GREB1* expression following E2 starvation (Fig 4). The distribution of initial promoter states among cells differed between steady-state (randomly sampled promoter states) and induction (all cells in the first OFF-state of the promoter cycle) conditions. The first 80 min of the time courses were not considered when fitting the steady-state data, which ensured an equilibration of the promoter state distribution.

### Global fitting

To pinpoint the effect of E2 stimulation on the promoter cycle, we simultaneously fitted multiple datasets, while assuming that most reactions are described by global, condition-independent parameter values (Fig 6). We fixed the model topology and allowed only one parameter at a time, namely initiation rate $k_{init}$ or OFF-time $t_{OFF}$, to vary locally, that is, with the experimental condition. To create a

start population of 1,000 particles for the global fitting, we filtered the results of the condition-specific fits (Fig 3) for overlap in their posterior distributions (see Appendix Supplementary Methods for details). The simulation of each experimental condition was compared to its corresponding dataset, and a total distance over all experimental datasets was calculated. As described above, parameter values were perturbed in each iteration using proposal distributions until no further improvement occurred.

### Data and software availability

The computer code to run simulations and model fits (Python code and IPython notebooks) can be found on the following GitHub resource: https://github.com/baumgast/gene_transcription_SMC_ABC

**Expanded View** for this article is available online.

## Acknowledgements

This study was supported by IMB's Flow Cytometry Core Facility, the use of its FACS Aria (DFG, INST 247645-1 FUGG) and LSR Fortessa (DFG, INST 247/646-1 FUGG) as well as supported by IMB's Microscopy Core Facility and the use of its AF7000 (DFG, INST 247/648-1 FUGG) and Opera Phenix (DFG, INST 247/845-1 FUGG) is gratefully acknowledged. We thank Helle Ulrich for providing access to the DeltaVision microscope (ERC AdG-323179). S.L., S.B., and C.F. were supported by the e:bio junior group program of the German Federal Ministry of Education and Research (BMBF, FKZ: 0316196). The work was also supported by the European ERASysBio+ Initiative Project Systems Approach to Gene Regulation Biology Through Nuclear Receptors (SYNERGY), Bundesministerium für Bildung und Forschung Grants ERASysBio+ P#134 to George Reid. The funders had no role in study design, data collection and analysis, decision to publish, or preparation of the manuscript.

## Author contributions

CF, SB, GR, and SL designed experiments, interpreted results, and wrote the manuscript. CF established the knock-in cell lines and performed imaging and all image analysis. SB implemented the SMC ABC algorithm and performed model fitting. CF and MK performed cloning, genotyping, and RT–qPCR experiments. DS performed high-content-imaging experiments. GR and SL obtained funding and supervised the project.

## Conflict of interest

The authors declare that they have no conflict of interest.

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
