## [Review Process File · Molecular Systems Biology]

Estrogen-dependent control and cell-to-cell variability of transcriptional bursting

Christoph Fritzscht, Stephan Baumgärtner, Monika Kuban, Daria Steinshorn, George Reid and Stefan Legewie

Review timeline:

Submission date:	18 April 2017
Editorial Decision:	20 June 2017
Revision received:	9 October 2017
Editorial Decision:	18 December 2017
Revision received:	16 January 2018
Accepted:	26 January 2018

Editor: Thomas Lemberger

Transaction Report:

1st Editorial Decision

20 June 2017

Thank you again for submitting your work to Molecular Systems Biology. I apologize again for the delay in getting back to you. In absence of any response of one of the initial reviewers, we were obliged to reassign new reviewers. We have now heard back from them. As you will see from the reports below, the referees find the topic of your study of potential interest. They raise, however, several concerns on your work, which should be convincingly addressed in a major revision of this work.

The reviewers raise a variety of points both on technical aspects and on the biological and functional levels. Without repeating all the points made in the reports below, two major issues that should be addressed are the following:

- the potential sources of extrinsic noise (eg cell cycle stage, cell volume) should be investigated
- the conclusions related to the action of HDAC inhibitor should be strengthened, in particular to ascertain the claim that mean expression and noise can be decoupled.

On a more editorial level, we would encourage to make the key datasets underlying this study available. Please see the data deposition section (<http://msb.embopress.org/authorguide#datadeposition>) in our instructions to authors.

We would also encourage you to include the source data for figure panels that show essential data, so that readers can download these data directly from the figure. Source data files are associated to individual panels of main figures. *Numerical data* should be provided as individual .xls files (including a tab describing the data) or csv or tab-delimited text files. *For 'blots' or microscopy*, uncropped images should be submitted. For *network visualization*, Cytoscape session files, if available, can be supplied. The files should be labeled as "Source Data for Figure 1A" etc. Source

Data for Expanded View and Appendix figures should be uploaded as a single ZIP file containing all the Source Data for Expanded View and Appendix content. (Additional information on source data is available in the "Guide for Authors" section at <http://msb.embopress.org/authorguide#sourcedata>).

If you feel you can satisfactorily deal with these points and those listed by the referees, you may wish to submit a revised version of your manuscript. Please attach a covering letter giving details of the way in which you have handled each of the points raised by the referees. A revised manuscript will be once again subject to review and you probably understand that we can give you no guarantee at this stage that the eventual outcome will be favorable.

REVIEWER REPORTS

Reviewer #1:

In this very strong paper, the authors combine live imaging and mathematical modeling to understand transcriptional behavior (in particular bursts of transcription) at the single-allele level. They use PP7 live imaging reporters to measure transcription site intensity upon estradiol induction, measuring intensity and duration over time in a large number of cells. They construct and analyze models to understand the implications of the results. Specifically, they construct various models of transcription with different numbers of states, finding that their data is well fit by the most parsimonious two-stage model. By constructing and analyzing a dual-reporter, they found the two alleles did not correlate in their firing, although there was still trans extrinsic variability leading to correlated differences in total output. They end with some analysis of HDAC inhibition.

Overall, I found this to be a really nice paper. The level of technical rigor is very high, and the results will be of great interest to the transcription community. The live imaging approach here is very well done, and in particular the dual-reporter results were very nice. The figures were also generally quite clear and informative.

Here are a few points to consider:

I think the authors' claims regarding the effects of HDAC inhibition are a bit stronger than the data support. For instance "Interestingly, we observed that butyrate lowers expression without increasing the noise (Fig 7A), implying that noise and mean can be controlled independently." There is basically just one perturbation for which it happens to be the case that the mean changed and the noise level did not, but we don't know what the curve looks like in between the control and the experimental condition, so it's difficult to say that it's not that it just a coincidence at that particular dose, not to mention the fact that HDAC inhibitors can have many non-specific effects. I would remove claims about being able to tune these two parameters independently unless there is stronger evidence for the claims.

A key point of the paper is that just a 2 state (or even 10-state) model was insufficient to model the observed data, the authors had to add in cell-to-cell variability in K_{elong} and K_{init} to get a good fit. It would be really helpful to see the fits of models with no cell-to-cell variability vs the favored model with the varying parameters to see exactly how they differ.

In Figure 3B it appears that only data is presented but not the model fit.

In Figure 4 the mean start time of the model vs data is somewhat different, which the authors discussed, but the variability between start times is MUCH greater in the data compared to the model (especially at 10pM E2). Do the authors have a hypothesis for why this is?

The authors use the metric of summed intensity for total RNA output. Is there a citation for validation of this fact? While there is a plausible relationship, there could be many potential confounders, like signal detection thresholds and so forth. I think it's at least worth discussing these caveats and how they might affect the interpretations.

The correlation in Fig. 5D is not so close to the model at the low end. Can the authors speculate as to why this may be the case?

Arjun Raj

Reviewer #3:

The authors analyzed mRNA production from the GREB1 locus in response to estrogen stimulus using a PP7 live-cell reporter system. They used CRISPR/Cas9 to integrate stem loop sequences into the endogenous GREB1 locus. They tracked the production of mRNAs over a period of several hours and quantified characteristics of mRNA bursting as a function of stimulus concentration and time. They next used a computational approach to identify plausible models of promoter activation and estimate kinetic parameters governing gene activation, finding that a two-state model recapitulates the major features of their experimental data, including E2 dose-dependent behavior. To determine whether variations in expression of GREB1 were due to intrinsic or extrinsic noise, they developed a system to simultaneously monitor mRNA production from both alleles of GREB1, finding that most fluctuations appear to arise from extrinsic factors. Finally, using butyrate (an HDAC inhibitor) they showed that intrinsic noise can be decoupled from mean RNA expression.

Overall, the work described in the manuscript is technically well done and provides detailed characterization of stochastic gene expression from the GREB1 locus. However, there appears to be a lack of major advances to the field from these experiments. A two-state model of promoter activation, as supported by this work, has been proposed and supported by several other studies of stochastic gene expression in single cells, as noted by the authors in this manuscript. The variation in burst frequency with increasing stimulus supports several previous studies of native transcripts at fixed time-points and time-course data from synthetic systems, so these studies of an endogenous locus are only incremental and not particularly novel. In general, the findings are rather vague and speculative in terms of biological mechanism and significance, with minimal discussion of connection to activities of the estrogen response system. Enthusiasm for this work would be increased with additional effort regarding more direct connections to the biology of the estrogen response system - both through experiments and discussion.

Major comments:

1. There is insufficient data to support the calibration of the fluorescence intensity measurements to exact RNA levels. How can one be sure that a single spot at full excitation intensity corresponds to a single RNA? Are variations in the intensity of individual transcriptional sites due to variations in the number of RNAs as claimed, or due to variations in the amount of fluorescent protein binding to the stem loops? I would suggest either renaming the axes in several figures in terms of relative changes in spot fluorescence instead of exact RNA levels (such as was done for smRNA FISH studies such as in Bartman, Hsu, Hsiung, Raj, and Blobel, *Mol Cell* 2016) (and which wouldn't detract from the conclusions of the manuscript), or alternatively perform quantification of stem loop fluorescent protein saturation (as done in Golding, Paulsson, Zawilski, and Cox, *Cell* 2005) and validation of RNA counts via a complementary approach such as RNA FISH (as performed by Lenstra, Coulon, Chow, and Larson, *Mol Cell* 2015) to better support the claims of precise RNA levels.
2. Analysis of the dual allele system suggests that extrinsic noise is the major source of fluctuations in GREB1 expression, but there is little done experimentally to follow up with this very general result. What sources of extrinsic noise are important? The data that currently exist may already be able to address some aspects of this question. For example, how does cell size impact fluctuations? How long do correlations in expression between sister cells persist after mitosis? One might expect that sister cells have higher correlation in extrinsic noise compared with two randomly selected cells.
3. Related to comment 2, estrogen receptor alpha is cell cycle regulated, so one might expect some cell cycle dependency of fluctuations in gene expression. How does cell cycle impact RNA production in the dual reporter system? This could be addressed by making use of cell cycle synchronization techniques (for example, mitotic inhibition and release) or a live-cell cell cycle marker (Fucci system? Not sure if enough fluorescence channels are available). Results from such

analysis may provide greater insight into biological function.

4. The claims regarding the roles of HDACs in affecting burstiness are a bit premature given that the authors have only looked at one inhibitor, butyrate, which has other biological activities. I would suggest corroborating their results with other HDAC inhibitors (preferably from other inhibitor classes) to strengthen the claim.

Minor comments:

1. Figure 6A and 7C - what is "distance [triangle] better"? It is not clear what is actually being plotted here.

2. On page 14, at the end of the Results section, it is stated "According to analytical calculations, uncoupling of noise...". If these calculations are newly performed by the authors, they should show the calculations. If they are calculations from previous studies, they should provide citations.

3. Better rationale for the use of HDAC inhibitors would improve the manuscript. In its current form, this line of experiments is described with insufficient justification.

Reviewer #4:

Stochastic gene expression is a major source of cellular heterogeneity. In the present manuscript, Fritzsche et al. investigate the dynamics of transcription at the endogenous estrogen-responsive GREB1 locus in a mammalian cell line. Using the PP7 system for monitoring (nascent) RNA in living cells, they demonstrate that estrogen modulates the frequency of transcriptional bursts, specifically the duration of the transcriptionally inactive state of the promoter, in a dose-dependent manner. Combining quantitative data from hundreds of cells with stochastic model simulations, they provide evidence that burst sizes are further affected by the global state of the cell, leading to increased cellular heterogeneity. Finally, the authors use HDAC inhibitors to independently modulate burst sizes, but not frequencies, which may provide means to fine-tune transcriptional output and associated noise levels in a given cell.

The manuscript represents, in my opinion, a fine example of a study combining highly quantitative experimental data with informative mathematical modelling. While understanding the origin and consequences of cellular heterogeneity is a highly relevant topic, stochastic gene expression has been studied in living cells for several years now and a variety of promoter models have been suggested. The experimental advantage of the present study is the use of time-resolved dose-dependent data from a single inducible locus in its natural chromatin environment. The modelling approach, specifically the combined model selection and fitting procedure, is interesting and allows estimating contributions from different noise sources, which is not trivial for endogenous loci. I therefore believe that the study is suitable for publication in MSB if the following points are addressed:

- 1) The authors should add controls by orthogonal methods to validate their experimental system. It would be helpful if they would
 - a) provide evidence that the imaged foci are indeed representing a GREB1 start site and indicate how many GREB1 alleles are present in MCF7 cells, which are tri- to tetraploid.
 - b) determine in more detail to which extent the insertion of PP7 does disturb transcription at the GREB1 locus - the RT-qPCR analysis in Fig. EV1B indicated that fold change at saturating estrogen levels is reduced by about 30%.
 - c) validate key measurements at the single cell level using a method like single-molecule FISH - this includes heterogeneity of mRNAs produced at different estrogen levels (e.g. Fig 2A) and heterogeneous induction of GREB1 after starvation (Fig. 4A)
- 2) A main conclusion of the paper is that global (extrinsic) effects dominate cell-to-cell variability, which is mainly based on a cell line with two tagged GREB1 alleles. There are several points regarding this:
 - a) Why did the authors use different insertion sites for both alleles? They should be able to tag the same site by using an sgRNA whose binding site is lost upon insertion in the first allele.

- b) The authors claim that model simulations produce closely matching correlations between RNA production rates from different alleles (page 12, Fig. 5D-E). However, the calculated correlation coefficients are significantly different, and also visually, the simulation seems to produce more correlated results than present in the data. Doesn't this indicate that the contribution of extrinsic noise is overestimated in the model? As a minor point, it would be fair to also plot data points from random pairs in Fig. 5D in comparison to the model without extrinsic noise in 5F.
- c) Recent papers from the Raj, Pelkmans and Itzkovitz labs have shown how global effectors such as cell cycle, cellular volume and nuclear buffering modulate gene expression. The authors only briefly mention these in the discussion. It would strengthen their conclusions if the authors would experimentally test sources of extrinsic noise in their system, for example by extracting some of these features (at least cell volume and cell cycle) from images and determining to which extent they can explain the "extrinsic" noise observed in the data.
- d) The authors assume extrinsic noise to be stable during the observation period (13h), which is about half the length of a cell cycle. Can they comment why that is the case? Usual sources of extrinsic noise, such as the number of polymerases may change during that time.

3) The authors validate the identified two-state model using estrogen-mediated induction of gene expression after starvation.

- a) Why do the authors use a ten-fold excess in E2 here compared to previous experiments (1000pM vs 100pM)? RT-qPCR (Fig. EV1B) seems to indicate that this concentration is well above saturation.
- b) It is not obvious from the heatmap shown that the two-state model is indeed superior to multi-state models. The delay before induction at lower doses is for example better reflected by a multi-state model. In addition, it is not obvious to which extent gene expression is more "regular". The authors could provide a more quantitative analysis here to convince readers.
- c) The authors explain the lag phase upon stimulation by signaling and chromatin modification processes. The explanation is not entirely convincing. How does this for example fit with the proposed two-state model? Would there be difference between expression at steady-state and during induction? Is the two-state model then still valid to describe the experimental data? And why is this lag time heterogeneous and decreasing with dose?

Minor points:

- The authors should explain why they use a time interval of 3 minutes for imaging. Is this sufficient to detect all bursts, specifically at lower estrogen doses? In other systems, faster transcriptional bursts have been demonstrated.
- Calibration of transcriptional start sites is based on spots only detected under very high light exposure, which are assumed to be single transcripts. This assumption should be validated experimentally. Furthermore, the authors then use a linear fit to derive a normalisation factor for quantifying the number of mRNAs at the start site for low illumination conditions. This assumes that experimental noise such as bleaching is also linear. Is there evidence for that? Wouldn't it be possible to calibrate the number of RNAs using another method such as smFISH?
- The identification of inactive start sites for low concentrations remains unclear, as they all have intensity values which are in the background range (see Fig 2D).
- It remains unclear where the number of 150 elongating polymerases on the body of the gene comes from (page 6)
- High-content imaging of fixed cells was performed by using a 20x objective to acquire 22 z-section 1.2µm apart. How can the authors ensure that all start-sites can be captured using these conditions which deviate from the optical settings usually used with the MS2/PP7 system?
- The authors claim that a model without extrinsic noise is not able to fit cell-to-cell variability. Can they provide evidence for this?
- How is residence time of transcripts at the GREB1 locus (as 30 min) measured/estimated? A single transcript (166448bps) could take between 110min (25 bp/sec PolII) and 37 min (75 bp/sec PolII) to be produced, depending on the range of published numbers for PolII transcription rates.
- On page 7, the authors state that varying amounts of RNA produced is an indication for stable extrinsic fluctuations. Isn't variability in RNA production a hallmark of all stochastic gene expression, independent of the noise source? Moreover, the underlying data only originates from one allele, making conclusions about intrinsic and extrinsic noise difficult. The authors should rephrase this section and rather focus on the data analysis / model simulations first before drawing conclusions about noise sources.
- On page 10, first line, Fig. 2E, not 2F should be referenced.

Reviewer #1:

In this very strong paper, the authors combine live imaging and mathematical modeling to understand transcriptional behavior (in particular bursts of transcription) at the single-allele level. They use PP7 live imaging reporters to measure transcription site intensity upon estradiol induction, measuring intensity and duration over time in a large number of cells. They construct and analyze models to understand the implications of the results. Specifically, they construct various models of transcription with different numbers of states, finding that their data is well fit by the most parsimonious two-stage model. By constructing and analyzing a dual-reporter, they found the two alleles did not correlate in their firing, although there was still trans extrinsic variability leading to correlated differences in total output. They end with some analysis of HDAC inhibition.

Overall, I found this to be a really nice paper. The level of technical rigor is very high, and the results will be of great interest to the transcription community. The live imaging approach here is very well done, and in particular the dual-reporter results were very nice. The figures were also generally quite clear and informative.

We thank the reviewer for his support and constructive criticism!

Here are a few points to consider:

1. I think the authors' claims regarding the effects of HDAC inhibition are a bit stronger than the data support. For instance "Interestingly, we observed that butyrate lowers expression without increasing the noise (Fig 7A), implying that noise and mean can be controlled independently." There is basically just one perturbation for which it happens to be the case that the mean changed and the noise level did not, but we don't know what the curve looks like in between the control and the experimental condition, so it's difficult to say that it's not that it just a coincidence at that particular dose, not to mention the fact that HDAC inhibitors can have many non-specific effects. I would remove claims about being able to tune these two parameters independently unless there is stronger evidence for the claims.

We initially addressed this issue by repeating the HDAC treatment and recorded nascent transcription time courses at an additional higher dose of NaBu (4 mM), which almost completely suppressed transcriptional activity. Consistent with our previous claims, we found that this treatment results in a lower noise when compared to the same mean expression achieved by estrogen titration (new Fig 7D), though under these conditions, the noise was no longer identical to the DMSO control, as we had observed with 2.5 mM NaBu, but slightly shifted upwards. This suggests that—at higher doses—NaBu treatment might not only affect the burst size, but also the burst frequency, possibly because the burst size becomes so low that occasionally no RNAs are produced from a burst. These effects on burst kinetics are consistent with model fits to the inhibitor and control data in which NaBu shifts the posterior distribution of the burst size at low doses, while additionally affecting the posterior distribution of burst-frequency (OFF-time) at higher doses (Fig 7E).

To further substantiate that NaBu regulates the burst size at low doses, we independently performed a dense titration of NaBu concentration and quantified transcription sites in fixed cells as a global intensity histogram (Fig EV5C). As expected, we found that low doses of NaBu primarily affect the intensity of the transcription sites (x-position of histogram), but not their frequency (area of the peak). In contrast, estrogen titration induced an immediate shift in the transcription site frequency, an effect that we also observed at higher doses of NaBu. Taken together, these data support that NaBu has a strong effect on the burst sizes, resulting in lower noise, as compared to frequency modulation by estrogen.

Finally, we performed similar analyses using PFII, a bromodomain inhibitor, which blocks the recognition of acetyl-residues by reader proteins, and also downregulated GREB1 transcription. Interestingly, in live-cell experiments, we observed the same trend of lower noise for a given mean,

when compared to estrogen titration (see Appendix Fig S9 and response to comment 4 of reviewer 3).

We added these findings to the revised manuscript (section “Burst properties and noise behavior are altered through protein acetylation”, pages 15-16).

2. A key point of the paper is that just a 2 state (or even 10-state) model was insufficient to model the observed data, the authors had to add in cell-to-cell variability in K_{elong} and K_{init} to get a good fit. It would be really helpful to see the fits of models with no cell-to-cell variability vs the favored model with the varying parameters to see exactly how they differ.

As suggested by the reviewer, we calibrated a two-state model without cell-to-cell variability to the data, assuming distinct parameters at different estrogen concentration concentrations (similar to previous Fig 3). For all estrogen concentrations, we found higher final distance values after convergence of the fitting algorithm, compared to the best fitting model that includes cell-to-cell variability (see Fig EV3G). Inspection of the fitting results revealed that a model without cell-to-cell variability cannot explain the smoothly decaying mean autocorrelation curve which we observed in the experimental data (see Appendix Fig S5). This is probably due to the fact that polymerase elongation, which partly determines the autocorrelation function, is too homogeneous in the absence of cell-to-cell variability.

To evaluate the necessity of a combined source of extrinsic noise in elongation and initiation rates, we fitted models with either of both sources to the 10 pM E2 dataset. We found that including cell-to-cell variability of the elongation rate alone provided a substantial improvement in quality of the fit. Including the initiation rate, however, improved the fit even further (see Fig EV3G). Thus, we conclude that the model with a combined source of extrinsic noise comprising of the elongation and initiation rate generates the best fitting from the set of possible models, supporting the model selection results shown in Fig 3C.

We added the sentence “By separately fitting two-state models with and without extrinsic noise sources to the data, we validated that extrinsic fluctuations in both the initiation and elongation rates are necessary to describe the experimental observations across a range of E2 concentrations” to the revised manuscript (page 11).

3. In Figure 3B it appears that only data is presented but not the model fit.

We believe that there might have been a problem with the pdf file or the viewer when displaying many semi-transparent lines. The Appendix Fig S4 contains the same information (although for the full dataset) but was converted to pixel graphics for easier viewing.

4. In Figure 4 the mean start time of the model vs data is somewhat different, which the authors discussed, but the variability between start times is MUCH greater in the data compared to the model (especially at 10pM E2). Do the authors have a hypothesis for why this is?

We agree that the mean and standard deviation of the response times were different between model and data, making it difficult to compare the two, and to evaluate the advantage of the two-state model over more complex topologies (see also major comment 3b by reviewer 4). To address this issue, we quantified the response time heterogeneity using the coefficient of variation ($CV = \text{standard deviation}/\text{mean}$). For an exponential distribution, i.e., a simple two-state model with a single step for gene reactivation, analytical calculations predict a $CV = 1$, whereas lower CVs are expected for multistep processes. In fact, when extracting single-cell response times from the synchronized data (as described in the revised Methods section), we observed $CV = 1.09$ for 1000 pM and $CV = 0.85$ for 10 pM estrogen (horizontal lines in Fig 4C), which is close to the theoretical prediction. The same response time extraction algorithm was applied to simulated synchronization time courses, in which the parameters from fits to 1000 pM continuous estrogen stimulation (Fig EV3E) were taken. As expected, we again observe more heterogeneous response times for small models (1 ON, 1 OFF state) when compared to larger promoter cycles (1 ON, 9 OFF states) (Fig EV4A), further supporting the qualitative agreement of small models and data.

Given that the quantitative comparison of stochastic simulations and experiments was complicated by very different mean response times (see also comment 3b by reviewer 4), we decided to change the presentation of model-data comparison under synchronization conditions. Specifically, we assumed that estrogen starvation could alter the internal state of the cell and therefore argued that the parameter values under synchronization conditions may be distinct from the ones in the previously calibrated model (Fig EV3E). Hence, we applied the previously established fitting algorithm (Fig 3) to the synchronization experiment data, and found that the two-state (1-1-5) model provides a quantitatively better fit than the ten-state (1-9-5) model for both induction conditions (new Fig EV4C). Using these 1-1-5 and 1-9-5 particles fitted to the synchronization experiment, we compared the simulated response time distributions to ones of the corresponding synchronization data (new Fig 4C). In this context, please note that the response time distributions were not used for fitting (but only the features in Fig 3B). In line with our hypothesis of a small model, we found that the fitted 1-1-5 particles recapitulated the experimentally observed response time CVs better than the 1-9-5 particles (Fig 4C). Hence, we believe that the variability of response times is similar for model and data, and therefore respectfully disagree with the reviewer. Our findings support our initial claim that the synchronization experiment can be better described by model topologies with few promoter activation steps.

We added this information to the section “A simple promoter model is recapitulated in induction experiments” on pages 11-12, and rephrased our conclusions concerning response time mean and heterogeneity.

5. The authors use the metric of summed intensity for total RNA output. Is there a citation for validation of this fact? While there is a plausible relationship, there could be many potential confounders, like signal detection thresholds and so forth. I think it's at least worth discussing these caveats and how they might affect the interpretations.

We are not aware of a citation using the summed intensity as total RNA output.

Notably, the summed intensity depends on two main features, the total number of RNAs produced in the time frame of the experiment and the dwell time of each transcript. To estimate a total number of RNAs per cell, we divided the summed intensity by the mean dwell time of the GREB1 transcript (as estimated from the gene length and PolIII speed). As an independent confirmation, we calculated the number of RNAs per cell by summing up all burst sizes, as directly extracted from the slopes of the time courses (described in the initial manuscript), and obtained similar numbers (not shown). To clarify the relation between summed intensities and total RNAs, we now mention the cell-to-cell variability of the summed intensity in the revised results (section “The productivity of GREB1 RNA synthesis exhibits considerable cell-to-cell variability”, page 7), and explain how we calculated the total number of RNAs from this.

We agree that signal detection thresholds may affect the results in quantitative terms. To analyze this, we calculated the summed intensity of the background signal, and found an average total RNA output of 12 RNAs in these background regions. We subtracted this value from the trajectories prior to noise calculations and achieved a more accurate inverse noise-mean scaling (new Fig 7A). We added this to the revised Methods.

6. The correlation in Fig. 5D is not so close to the model at the low end. Can the authors speculate as to why this may be the case?

We agree with the reviewer that there was an apparent mismatch between model and experiment, also in the overall Pearson Correlation coefficient (PCC) between allele 1 and 2. See also major comment 2b by reviewer 4, which we also address with this response.

There are two potential reasons for the discrepancy between allele correlations in model and data: (i) uncertainty in the value of the PCC for two distributions of limited sample size. (ii) simulated PCCs may be distinct for each particle in the posterior distribution. We first addressed the latter point and calculated allele correlations for all particles of the posterior distribution at 10 pM estrogen (median PCC = 0.8). We observed a large spread, with some particles showing PCC close to 1, while others fulfilled PCC \approx 0.5 (Appendix Fig S8, left boxplot). This suggests that the degree of extrinsic noise shows pronounced variation between particles, and can indeed closely match the

experimentally observed correlation for certain particles (PCC = 0.54; Appendix Fig S8, blue dashed line). To further assess the effect of statistical variation arising from limited sample sizes, we picked one particle and performed repeated simulations and PCC calculations. This particle had a median PCC = 0.70, and the PCC distribution spread out from 0.46 to 0.83 (5 and 95 percentiles), suggesting that the statistical variation for each particle was limited (Appendix Fig S8, second boxplot).

In the revised paper, we decided to show the correlation of the latter particle to show a better match between model and experiment in terms of PCC (Fig 5F), and also show the variability of the PCC due to particle choice and sample size (Appendix Fig S8). The larger spread at the lower end of the distribution of the data compared to the model, however, persists. We believe that there is either a data acquisition issue at low expression levels, or that additional allele-intrinsic (cis-acting) variability exists, e.g. due to differences at the chromatin level.

Given that the discrepancy only occurs at a few data points, we decided not to discuss the latter point in the revised manuscript due to length restrictions and only added Appendix Fig S8, which contains the uncertainty analysis for the sister allele correlation.

Reviewer #3:

The authors analyzed mRNA production from the GREB1 locus in response to estrogen stimulus using a PP7 live-cell reporter system. They used CRISPR/Cas9 to integrate stem loop sequences into the endogenous GREB1 locus. They tracked the production of mRNAs over a period of several hours and quantified characteristics of mRNA bursting as a function of stimulus concentration and time. They next used a computational approach to identify plausible models of promoter activation and estimate kinetic parameters governing gene activation, finding that a two-state model recapitulates the major features of their experimental data, including E2 dose-dependent behavior. To determine whether variations in expression of GREB1 were due to intrinsic or extrinsic noise, they developed a system to simultaneously monitor mRNA production from both alleles of GREB1, finding that most fluctuations appear to arise from extrinsic factors. Finally, using butyrate (an HDAC inhibitor) they showed that intrinsic noise can be decoupled from mean RNA expression.

Overall, the work described in the manuscript is technically well done and provides detailed characterization of stochastic gene expression from the GREB1 locus. However, there appears to be a lack of major advances to the field from these experiments. A two-state model of promoter activation, as supported by this work, has been proposed and supported by several other studies of stochastic gene expression in single cells, as noted by the authors in this manuscript. The variation in burst frequency with increasing stimulus supports several previous studies of native transcripts at fixed time-points and time-course data from synthetic systems, so these studies of an endogenous locus are only incremental and not particularly novel. In general, the findings are rather vague and speculative in terms of biological mechanism and significance, with minimal discussion of connection to activities of the estrogen response system. Enthusiasm for this work would be increased with additional effort regarding more direct connections to the biology of the estrogen response system - both through experiments and discussion.

We thank the reviewer for his/her comments, which have helped us to improve our manuscript.

Major comments:

1. There is insufficient data to support the calibration of the fluorescence intensity measurements to exact RNA levels. How can one be sure that a single spot at full excitation intensity corresponds to a single RNA? Are variations in the intensity of individual transcriptional sites due to variations in the number of RNAs as claimed, or due to variations in the amount of fluorescent protein binding to the stem loops? I would suggest either renaming the axes in several figures in terms of relative changes in spot fluorescence instead of exact RNA levels (such as was done for smRNA FISH studies such as in Bartman, Hsu, Hsiung, Raj, and Blobel, Mol Cell 2016) (and which wouldn't detract from the conclusions of the manuscript), or alternatively perform quantification of stem loop fluorescent protein saturation (as done in Golding, Paulsson, Zawilski, and Cox, Cell 2005) and validation of RNA counts via a complementary approach such as RNA FISH (as performed by Lenstra, Coulon, Chow, and Larson, Mol Cell 2015) to better support the claims of precise RNA levels.

To address this issue, we performed smFISH using a combination of intronic and exonic GREB1 probes to distinguish nascent and mature transcripts. Using the exonic probes, we could quantify single transcripts in the cytoplasm, and found that they had a narrow intensity distribution, which made us confident that we can indeed identify single RNA molecules. By comparing these intensities to bulky transcription sites (as identified by co-localization with signal from intronic probes, which primarily label nascent transcripts), we estimated that transcription sites contain up to ~150 nascent RNA molecules (Fig EV2D). This number is in agreement with the one derived by live-cell imaging using the PP7 system. More quantitatively, we matched the intensity distributions of transcription sites from smRNA FISH and live-cell imaging (as performed by Lenstra et al., Mol Cell, 2015) and observed excellent agreement of the two calibration methods (Fig EV1F and compare to EV1H). Interestingly, the number of nascent transcripts as determined from smRNA FISH images were very similar for endogenous and PP7-labeled GREB1 loci (the latter identified

using the GFP channel) (Fig EV1G), suggesting that the RNA numbers we obtain are physiological and not perturbed by the reporter system.

Concerning stem loop saturation, we analyzed if GFP expression level in a cell affects the intensity of a single RNA spot as measured by PP7 fluorescence at high excitation energies. We did not observe a significant correlation between spot intensity and GFP expression level, suggesting that the stem loops were saturated with PCP-GFP protein at all expression levels (see below).

Taken together, these findings suggest that we can indeed quantify precise RNA numbers. We added the sentence “This quantification was independently confirmed through exonic smRNA FISH (Fig EV1F)” to the main text (page 6, section “Digital modulation of GREB1 transcription by estrogen”).

2. Analysis of the dual allele system suggests that extrinsic noise is the major source of fluctuations in GREB1 expression, but there is little done experimentally to follow up with this very general result. What sources of extrinsic noise are important? The data that currently exist may already be able to address some aspects of this question. For example, how does cell size impact fluctuations? How long do correlations in expression between sister cells persist after mitosis? One might expect that sister cells have higher correlation in extrinsic noise compared with two randomly selected cells.

3. Related to comment 2, estrogen receptor alpha is cell cycle regulated, so one might expect some cell cycle dependency of fluctuations in gene expression. How does cell cycle impact RNA production in the dual reporter system? This could be addressed by making use of cell cycle synchronization techniques (for example, mitotic inhibition and release) or a live-cell cell cycle marker (Fucci system? Not sure if enough fluorescence channels are available). Results from such analysis may provide greater insight into biological function.

We agree with the reviewer that we did not provide a molecular basis for extrinsic fluctuations. We therefore further analyzed our imaging data to address this point, and extracted several morphological features including local cell density, nuclear area and shape, and total cell area. We then related these features to the total RNA output as a measure of extrinsic noise, but did not observe any strong correlation of transcriptional activity with any of these features. As a trend, we observe that cells with higher transcriptional activity tend to exhibit higher nuclear and cytoplasmic areas (Appendix Fig S2). This finding agrees with previous studies showing that the cell volume (that is presumably correlated with these areas) determines transcriptional output (Padovan-Merhar et al., Mol Cell, 2015; Kempe et al., Mol Biol Cell, 2015), but in our hands this relationship is weak at best, as also reported by Padovan-Merhar et al., who described low volume-dependency for cell type specific genes. We further employed a linear regression approach to investigate whether combinations of morphological features are better predictors of transcriptional outcome, but again did not find very strong predictive power (Appendix Fig S2).

We believe that the cell cycle stage can be excluded as a major source of extrinsic noise in our imaging conditions, since we discarded cells that show two transcription sites from a replicated allele at any time point during the observation period. Hence, the cells we analyze never pass through S, G2 or M, and are thus restricted to the G0/G1 phases of the cell cycle. We can nevertheless image most of the cells in the population over 12 hours, because MCF7 cells exhibit a very slow cell cycle time of 30 h, with G1 being the longest phase of about 17 h (Dalvai et al., PLoS One, 2010). We further believe that substantial transcription heterogeneity between G0 and G1, or within G1 (early vs. late) can be excluded: in the synchronization experiment, we trigger starved cells to exit G0 and to simultaneously enter the G1 phase. Despite these cells being in a similar cell

cycle stage, we observe substantial variation in transcriptional output, with some cells not or only weakly responding throughout the whole observation period (Fig 4A).

As suggested by the reviewer, we further analyzed daughter cells in a long movie in which we imaged transcription sites over 25 hours (Appendix Fig S3). In line with stable extrinsic noise, we found that sister cells have a higher correlation of total RNA output in the first 6h after mitosis compared to randomly selected cells (Appendix Fig S3B). Even though sister cells were intrinsically cell cycle synchronized, we found a similar degree of cell-to-cell variability in total RNA output ($CV^2 = 0.16$, Appendix Fig S3A) as we had observed in unsynchronized cells ($CV^2 = 0.19$ for 100 pM dataset in Fig 2A), further suggesting that cell cycle effects negligibly contribute to extrinsic noise. We could not analyze how long correlations between sister cells persist after mitosis, because too few cells divided within the 25 hour imaging period (Appendix Fig S3A).

Taken together, these observations suggest that neither simple morphological features nor the cell cycle stage can fully explain the extrinsic fluctuations in nascent transcription. In line with the absence of strong cell cycle effects, it has been reported that ESR1 transcript levels drop by only 40 % in S phase (Dalvai et al., PLoS One, 2010) and are further buffered through long protein half-lives. In the revised discussion we speculate that stochastic protein concentration fluctuations in the general transcription machinery or in the estrogen signaling pathway may explain long-lasting transcriptional differences and propose possible experimental validation strategies.

We added a paragraph to the section “The productivity of GREB1 RNA synthesis exhibits considerable cell-to-cell variability” (page 8) in the revised manuscript and discuss the finding in the revised Discussion.

4. The claims regarding the roles of HDACs in affecting burstiness are a bit premature given that the authors have only looked at one inhibitor, butyrate, which has other biological activities. I would suggest corroborating their results with other HDAC inhibitors (preferably from other inhibitor classes) to strengthen the claim.

We agree with the reviewer that HDAC inhibitors have multiple biological activities, and that it is very difficult to disentangle the impact of chromatin regulatory events on gene expression noise from our experiments. However, we did not focus on the molecular mechanisms at the chromatin level, but rather asked whether GREB1 expression can be lowered without increasing noise in expression. We pointed this out more clearly in the revised manuscript, and toned down our claims concerning molecular mechanisms of these inhibitors.

We performed several additional experiments to address the reviewers concern that the conclusions are premature (see also point 1 by reviewer 1)

- 1) We repeated the HDAC inhibition with an additional higher dose of NaBu (4 mM) in live-cell imaging (Fig 7B-D), and performed a dense NaBu titration in fixed cell experiments (Fig EV5C, see response to comment 1 of reviewer 1). These analyses corroborated that NaBu has pronounced effects on the burst size and allows for noise-mean uncoupling.
- 2) We performed fixed-cell transcription site quantifications using TSA, an hydroxamic acid HDAC inhibitor, and found that it has very similar effects on GREB1 expression as the carboxylate NaBu, as it substantially lowers GREB1 expression (Fig EV5A).
- 3) To support our findings by independent inhibitor experiments, we added PFII, a bromodomain inhibitor, to cells to perturb the recognition of acetylated residues by reader proteins. PFII also lowered GREB1 expression, and tended to do so at a lower noise level when compared to a purely burst frequency-modulated estrogen titration (Appendix Fig S9). This supports our claim that protein acetylation regulates the burst size.

Taken together, these data suggest that the effects we see are generally related to protein acetylation and not caused by non-specific side effects of NaBu. We briefly discuss these findings in the revised manuscript in section “Burst properties and noise behavior are altered through protein acetylation” on pages 15-16.

Minor comments:

1. Figure 6A and 7C - what is "distance [triangle] better"? It is not clear what is actually being plotted here.

"Distance" describes the dissimilarity of stochastic simulations with the data and is the metric that is optimized using SMC ABC. A lower value for the distance indicates a better match between simulations and data, and hence, that the parameters used for the simulations are more likely. We renamed the corresponding axes.

2. On page 14, at the end of the Results section, it is stated "According to analytical calculations, uncoupling of noise...". If these calculations are newly performed by the authors, they should show the calculations. If they are calculations from previous studies, they should provide citations.

We added the corresponding citation of Singh et al., Biophys J, 2010.

3. Better rationale for the use of HDAC inhibitors would improve the manuscript. In its current form, this line of experiments is described with insufficient justification.

We added a more comprehensive motivation for the HDAC treatment. See page 15.

Reviewer #4:

Stochastic gene expression is a major source of cellular heterogeneity. In the present manuscript, Fritzscht et al. investigate the dynamics of transcription at the endogenous estrogen-responsive GREB1 locus in a mammalian cell line. Using the PP7 system for monitoring (nascent) RNA in living cells, they demonstrate that estrogen modulates the frequency of transcriptional bursts, specifically the duration of the transcriptionally inactive state of the promoter, in a dose-dependent manner. Combining quantitative data from hundreds of cells with stochastic model simulations, they provide evidence that burst sizes are further affected by the global state of the cell, leading to increased cellular heterogeneity. Finally, the authors use HDAC inhibitors to independently modulate burst sizes, but not frequencies, which may provide means to fine-tune transcriptional output and associated noise levels in a given cell.

The manuscript represents, in my opinion, a fine example of a study combining highly quantitative experimental data with informative mathematical modelling. While understanding the origin and consequences of cellular heterogeneity is a highly relevant topic, stochastic gene expression has been studied in living cells for several years now and a variety of promoter models have been suggested. The experimental advantage of the present study is the use of time-resolved dose-dependent data from a single inducible locus in its natural chromatin environment. The modelling approach, specifically the combined model selection and fitting procedure, is interesting and allows estimating contributions from different noise sources, which is not trivial for endogenous loci. I therefore believe that the study is suitable for publication in MSB if the following points are addressed:

Again, we appreciate the care, thoughtfulness and support of referee 4. Addressing his/her comments has substantially improved our revised manuscript.

1) The authors should add controls by orthogonal methods to validate their experimental system. It would be helpful if they would

a) provide evidence that the imaged foci are indeed representing a GREB1 start site and indicate how many GREB1 alleles are present in MCF7 cells, which are tri- to tetraploid.

To address this issue, we performed dual-color smFISH using a combination of intronic and exonic GREB1 probes (see reviewer 3, major comment 1), and simultaneously imaged the GFP channel to monitor PP7-transcription sites in the same fixed cell (see Fig EV1D). In the smFISH channels, we typically observed up to three high-intensity spots (sometimes occurring in duplicate when the locus

already replicated) that were overlapping for intronic and exonic probes, suggesting that MCF-7 cells harbor three GREB1 alleles. For one of these high-intensity smFISH spots (two for the dual allele cell line, see Appendix Fig S7), we observed an excellent overlap with PP7 transcription sites. Furthermore, we quantified spot intensities and observe a strong correlation of intronic, exonic, and GFP intensities (Fig EV1E). We added a corresponding overlay of GFP and smFISH channels, as well as the intensity quantification to Fig EV1 and report these findings in the main text when introducing the reporter system on page 5.

b) determine in more detail to which extent the insertion of PP7 does disturb transcription at the GREB1 locus - the RT-qPCR analysis in Fig. EV1B indicated that fold change at saturating estrogen levels is reduced by about 30%.

We agree that it is important to address how the insertion of PP7 sequences affects transcription. In Fig EV1B, we plotted the fold-change of GREB1 mRNA over basal, and observed that the PP7 allele tends to show reduced expression. However, one has to make a distinction between de novo RNA production (the main interest of this manuscript) and steady-state RNA levels. Only the latter one is measured by allele-specific RT-qPCR, but it critically depends on the stability of the target RNA. We suspect that the presence of PP7 stem-loops in the exon of the GREB1 mRNA leads to a destabilization of the mature RNA and it is this difference in stability that produces the change in mRNA levels between wildtype and knock-in allele. The RT-qPCR measurements were mainly performed to analyze the sensitivity towards E2. As both curves show comparable EC_{50} values, we concluded that the sensitivity is unaffected, and this conclusion still holds true, even if the maximum RNA levels are slightly different.

To address the question whether the knock-in affects RNA production itself, we performed smRNA FISH for GREB1 exons and quantified the intensities of transcription sites that overlapped with a spot in the GFP channel (knock-in alleles) and compared them with spots that do not co-localize with a GFP spot (i.e. wildtype alleles) (see Fig EV1G). If the PP7 sequences would not alter the de novo production of RNAs, the intensities from both alleles should be comparable. Indeed, we did not observe differences in spot intensities across five different concentrations of E2, confirming that nascent transcription is not disturbed by the presence of the PP7 cassette.

In the revised manuscript, we added the quantification of spot intensities (Fig EV1G) and mention that the absolute GREB1 mRNA levels may be slightly different (caption Fig EV1B), whereas the stimulus dependency is well preserved.

c) validate key measurements at the single cell level using a method like single-molecule FISH - this includes heterogeneity of mRNAs produced at different estrogen levels (e.g. Fig 2A) and heterogeneous induction of GREB1 after starvation (Fig. 4A)

To address this issue, we used smFISH to measure transcription site intensities. When comparing intensities of single RNA molecules to the intensities of these sites, we found that a transcription site contains up to 150 nascent RNAs, which is in excellent agreement with our estimate based on the PP7 reporter system. Please see response to reviewer 3, major comment 1 for more details.

Furthermore, we compared the bimodal intensity distributions of smFISH (exonic probes) and PP7 transcription sites at different estrogen concentrations (Fig EV2D) and found a good overlap, suggesting that nascent RNA noise levels are comparable when assessed by these complementary measurements. We did not analyze cytoplasmic mRNA levels and associated heterogeneity, because these measurements do not reflect well the intrinsic noise that we obtain by nascent RNA analysis. Steady state levels of cytoplasmic RNAs depend on factors such as mRNA export and stability. In combination with the reduced noise due to the presence of multiple (three) GREB1 alleles, this would obfuscate noise measurements and hinder comparability to our live-cell PP7 data.

Given that we had already validated the induction kinetics of GREB1 under synchronization conditions using independent RT-qPCR measurements, we decided to refrain from a further smFISH-based validation of heterogeneity in this experiment. We think that a quantitative validation of the heterogeneous induction kinetics would require snapshot measurements at many time points and believe that such analyses is beyond the scope of the current manuscript.

2) A main conclusion of the paper is that global (extrinsic) effects dominate cell-to-cell variability, which is mainly based on a cell line with two tagged GREB1 alleles. There are several points regarding this:

a) Why did the authors use different insertion sites for both alleles? They should be able to tag the same site by using an sgRNA whose binding site is lost upon insertion in the first allele.

We respectfully believe that there might have been a misunderstanding concerning the dual allele cell line, and would like to clarify that the two alleles in this cell line were identical, i.e., they both harbor PP7 sequences in intron 2, whereas exon 2 (labeled in the single allele cell line) remained unperturbed. Thus, the dual and single-allele were generated independently.

The main reason to target the intron in the dual allele cell line was cellular fitness. GREB1 has been reported to act as an important growth mediator in response to estrogen stimulation (Rae et al., Breast Cancer Res. Treat., 2005). Given that PP7 sequences may disturb RNA stability or translation (Urbanek et al., RNA Biol., 2014), we reasoned that cells would behave more naturally if PP7 sequences are spliced out.

We changed the following sentence on page 13 of the revised manuscript to make both points clearer for the reader: "To maintain protein function of the cell growth regulator GREB1, the PP7 sequences were delivered into intron 2 at two out of three GREB1 alleles, about 1 kb downstream of the above-described knock-in site within exon 2."

b) The authors claim that model simulations produce closely matching correlations between RNA production rates from different alleles (page 12, Fig. 5D-E). However, the calculated correlation coefficients are significantly different, and also visually, the simulation seems to produce more correlated results than present in the data. Doesn't this indicate that the contribution of extrinsic noise is overestimated in the model? As a minor point, it would be fair to also plot data points from random pairs in Fig. 5D in comparison to the model without extrinsic noise in 5F.

We address this point in our response to reviewer 1, comment 6. Please see there. Briefly, we assessed statistical variation in the correlation of sister allele simulations and added Fig S8 to the Appendix. As suggested, we also added random pairs to Fig 5D.

c) Recent papers from the Raj, Pelkmans and Itzkovitz labs have shown how global effectors such as cell cycle, cellular volume and nuclear buffering modulate gene expression. The authors only briefly mention these in the discussion. It would strengthen their conclusions if the authors would experimentally test sources of extrinsic noise in their system, for example by extracting some of these features (at least cell volume and cell cycle) from images and determining to which extent they can explain the "extrinsic" noise observed in the data.

We address this point in our response to reviewer 3, major comment 2. Please see there.

d) The authors assume extrinsic noise to be stable during the observation period (13h), which is about half the length of a cell cycle. Can they comment why that is the case? Usual sources of extrinsic noise, such as the number of polymerases may change during that time.

We agree that previous studies showed that heterogeneity in protein expression has a limited lifetime. Sigal et al. analyzed how long-lasting cell-to-cell variability in protein expression is and reported that high and low expressing cells mix with a half live of 24 to 60 hours, depending on the protein (Sigal et al., Nature 2006). Given that these numbers are longer than our observation period, we decided to choose a stable extrinsic noise source. In line with this assumption, we observed that the total RNA output within a cell is highly correlated between the first and the second half of our experiment (Appendix Fig S1A). In quantitative agreement, we find a similar correlation in simulations in which a stable extrinsic noise contribution was assumed (Appendix Fig S1B).

We added the sentence "Cell-to-cell variability is stable over time, as the RNA output during the first half of the movie correlates with the RNA output during the second half (Appendix Fig S1A)." in the revised results part on page 7 to motivate the assumption of a stable extrinsic noise source.

3) The authors validate the identified two-state model using estrogen-mediated induction of gene expression after starvation.

a) Why do the authors use a ten-fold excess in E2 here compared to previous experiments (1000pM vs 100pM)? RT-qPCR (Fig. EV1B) seems to indicate that this concentration is well above saturation.

In the starvation experiment, we aimed for a characterization of promoter switching from an OFF- to an ON state, and sought to exclude delays from the upstream estrogen signaling pathway as much as this is possible. From theoretical and experimental analyses, it has been concluded for signaling pathways that the kinetics of transcription factor activation is accelerated at higher doses of ligand when compared to low doses (e.g., Heinrich & Neel, Mol Cell, 2002; Salazar & Hoefler, JMB, 2003). Therefore, the use of a high estrogen concentration minimizes additional delays, while in our opinion having no disadvantages for our response time analyses. Other studies that induce transcription after estrogen starvation use an E2 concentration of 10 nM (Métivier et al., Cell, 2003; Perillo et al., Science, 2008; Sharp et al., JCS, 2006) or even 100 nM (Shang et al., Cell, 2000; Li et al., Nature, 2013), such that we believe that our 1 nM treatment, while still being saturating, is rectified. We decided not to discuss this point in the manuscript due to space limitations of the journal.

b) It is not obvious from the heatmap shown that the two-state model is indeed superior to multi-state models. The delay before induction at lower doses is for example better reflected by a multi-state model. In addition, it is not obvious to which extent gene expression is more "regular". The authors could provide a more quantitative analysis here to convince readers.

We address this point in our response to reviewer 1, comment 4 . Please see there.

c) The authors explain the lag phase upon stimulation by signaling and chromatin modification processes. The explanation is not entirely convincing. How does this for example fit with the proposed two-state model? Would there be difference between expression at steady-state and during induction? Is the two-state model then still valid to describe the experimental data? And why is this lag time heterogeneous and decreasing with dose?

We address this point in our response to reviewer 1, comment 4. Please see there. In brief, we show that this lag phase no longer needs to be assumed and that the response can be quantitatively described by the simple two-state model (assuming the bursting parameters to be slightly different in the starvation experiment). As we showed towards the end of our manuscript, estrogen changes the frequency of transcriptional bursts (i.e., the time required for gene reactivation). Hence, it is not surprising and in line with the two-state model that the lag time is decreasing with dose, and the stochastic switching in this model establishes the observed heterogeneity in the response times (see response to reviewer 1, comment 4).

Minor points:

- The authors should explain why the use a time interval of 3 minutes for imaging. Is this sufficient to detect all bursts, specifically at lower estrogen doses? In other systems, faster transcriptional burst have been demonstrate.

The gene structure of GREB1 and the location of the PP7 knock-in in the 5' UTR in exon 2 leads to an elongation time for the rest of the gene of ~30 minutes. Hence, transcripts that contribute to PP7 fluorescence accumulation are observable for much longer than the 3 minute imaging interval. We agree that the duration of a burst might be shorter than 3 minutes, but the signal of nascent transcripts is then easily picked up in the next frame of the movie and still lasts for ~10 frames. We changed the sentence on page 6 to incorporate this point: "[...] with an imaging interval of 3 minutes, which is well below the estimated ~30 minutes residence time of individual, nascent GREB1 RNAs at the locus (Fig 1E)."

- Calibration of transcriptional start sites is based on spots only detected under very high light exposure, which are assumed to be single transcripts. This assumption should be validated experimentally. Furthermore, the authors then use a linear fit to derive a normalisation factor for quantifying the number of mRNAs at the start site for low illumination conditions. This assumes that experimental noise such as bleaching is also linear. Is there evidence for that? Wouldn't it be possible to calibrate the number of RNAs using another method such as smFISH?

We addressed the calibration of fluorescence intensities through complementary quantitative smRNA FISH measurements. Please see our response to reviewer 3, major comment 1. The normalization factor to convert fluorescence intensities from maximum excitation to low excitation was derived from snapshot measurements at one time point during which bleaching is minimal. We therefore think that the assumption of a linear relationship holds.

- The identification of inactive start sites for low concentrations remains unclear, as they all have intensity values which are in the background range (see Fig 2D).

Transcription sites were tracked in the nucleus, and they were observed to be relatively immobile with respect to the nucleoplasm. Hence, as soon as a transcription site is visible at least once during a movie (and this occurs for most cells, even at low E2 concentrations), its position can be tracked. If no spot was detectable throughout the whole movie, a position close to the center was used. While this is unlikely to reflect the true position of the GREB1 gene, it provides a similar "background" intensity as other positions in the nucleus.

We added this information to the revised Methods section "Live-cell image analysis".

- It remains unclear where the number of 150 elongating polymerases on the body of the gene comes from (page 6)

The maximum fluorescence intensity that was observed throughout all movies is equal to 150 RNAs when converted using the intensity of a single RNA. During editing of Figures 2B and EV2B for the revised manuscript we realized a slight scaling issue in the trajectories of single cells at 100 and 1000 pM E2. It is now more obvious that a maximum value of 150 RNAs can be achieved.

- High-content imaging of fixed cells was performed by using a 20x objective to acquire 22 z-section 1.2µm apart. How can the authors ensure that all start-sites can be captured using these conditions which deviate from the optical settings usually used with the MS2/PP7 system?

The 20x water objective that was used during high-content imaging has an NA of 1.0. This gives rise to a depth of focus of 1.8 µm, which is indeed higher than for the high-NA objectives with a higher magnification that are usually used for live imaging with the PP7 system, but sufficient to resolve transcription sites. We are aware that we are not imaging at the optimal Nyquist sampling of ~0.8 µm, but with a distance of 1.2 µm we are still below the depth of focus such that all sites are captured. This is exemplified in the figure below, in which a bright (red) and a dim (blue) transcription site are shown as maximum intensity projections along the z- or the y-axis. Even the dim transcription site is visible in at least two z-slices.

We decided not to discuss this in the revised manuscript due to space limitations.

-The authors claim that a model without extrinsic noise is not able to fit cell-to-cell variability. Can they provide evidence for this?

We address this point in our response to reviewer 1, comment 2. Please see there.

-How is residence time of transcripts at the GREB1 locus (as 30 min) measured/estimated? A single transcript (166448bps) could take between 110min (25 bp/sec PolIII) and 37 min (75 bp/sec PolIII) to be produced, depending on the range of published numbers for PolIII transcription rates.

The longest GREB1 isoform (ENST00000381486.6) has a length of 108.672 bp, with 86.303 bp being transcribed after the PP7 sequences in exon 2. Because transcripts are only visualized once PolIII transcribed the PP7 sequences, only the ~86 kb contribute to the dwell time of a transcript. Our estimation of PolIII transcription rate is based on kinetic polymerase occupancy measurements along the gene after an estrogenic signal (wa Maina et al. 2014 and personal communication with Magnus Rattray). Furthermore, time-resolved RT-qPCR measurements for the exon-intron boundaries after exon 2 (close to the site of knock-in) and before exon 33 (at the end of the gene) after estrogen induction (Fig EV4B) revealed a lag time between both PCR products of about 25 minutes, in line with the estimated 30 minute residence time of transcripts. We added “[...] with the 30 minutes which a transcript is observable estimated from gene length and published PolIII elongation rates.” to the caption of Fig1B.

-On page 7, the authors state that varying amounts of RNA produced is an indication for stable extrinsic fluctuations. Isn't variability in RNA production a hallmark of all stochastic gene expression, independent of the noise source? Moreover, the underlying data only originates from one allele, making conclusions about intrinsic and extrinsic noise difficult. The authors should rephrase this section and rather focus on the data analysis / model simulations first before drawing conclusions about noise sources.

We agree that variability in RNA production is a hallmark of stochastic gene expression, and that there are two contributions to this phenomenon: (i) transcriptional bursting due to low molecule numbers at the promoter level (intrinsic noise); (ii) stable differences in transcriptional permissiveness (extrinsic noise). We think that the distinction between extrinsic and intrinsic noise is important, as the former dominates fluctuations on short time scales, whereas the latter predominates on longer time scales. We show this separation of time scales in the dual allele cell line, in which the two alleles are uncorrelated in snapshot measurements, but correlated when gene activity is averaged over time.

We continue to believe that the total amount of RNA that is being produced during our movies is a valid measure for extrinsic noise. By calculating the total RNA output over a period of 13 hours, the intrinsic noise component due to bursting is effectively “averaged out”, such that differences extrinsic noise can be estimated. We provide two lines of evidence supporting this claim: (i) we calculate the total RNA output in the 1st and 2nd half of the time courses, and find a pronounced correlation in both measures (Appendix Fig S1, see response to comment 2d), suggesting that a 12h period is sufficient to average out bursting effects; (ii) we performed stochastic simulations using calibrated model parameters, and found that the total number of RNAs during the observation period is less distinct between cells compared to the experimental data if we eliminate extrinsic noise sources from the model (i.e., bursting alone, on a realistic time scale, cannot explain these long-term variations). We added point (i) to the main text (page 7) to support our claims about extrinsic fluctuations, and further discuss the advantages and disadvantages of our experimental system to measure extrinsic noise at the level of nascent transcription in the Discussion (page 19).

-On page 10, first line, Fig. 2E, not 2F should be referenced.

We thank the reviewer for pointing out this inaccuracy and corrected it in the manuscript.

Thank you again for submitting your work to Molecular Systems Biology. We have now heard back from the two referees who accepted to evaluate the revised study. As you will see, the referees are now supportive and I am pleased to inform you that we will be able to accept your paper for

publication in Molecular Systems Biology pending the following minor amendments:

- reviewer #4 asks for some final clarifications
- we would need ORCID identifiers for both corresponding authors (currently missing for G. Reid)
- Dataset EV1-EV6 need to be explicitly called out from the main text
- Reference list is 10 authors + et al, but should be 20 authors + et al
- We appreciate that you included the movie legends in the Appendix. We would be grateful if you could also include the movie legends as text only files (eg labeled README) and zip the respective legend with its movie file.
- we would suggest to include a Data and software availability section placed after Materials and Methods that include the following:

Data and software availability

The computer code to run simulations and model fits can be found on the following resource:

- Python code and IPython notebooks: GitHub
https://github.com/baumgast/gene_transcription_SMC_ABC

REVIEWER REPORTS

Reviewer #1:

The authors have done a good job responding to my comments, and the manuscript looks great.
Arjun Raj

Reviewer #4:

In general, the authors have very well addressed my comments to the initial manuscript, specifically by adding independent validation through smFISH.

In their response to point 2.c (reviewer 3 comment 2), they mention that "sister cells have a higher correlation of total RNA output in the first 6h after mitosis compared to randomly selected cells". I may have missed something, but as far as I can see, only the correlation for sister cells is shown in the corresponding figure S3B. If this is true, maybe the authors can add the data for randomly selected cells as comparison.

In response to the same issue, the authors analyze the correlation of transcriptional outcome to combinations of morphological features and claim that they "did not find very strong predictive power". In the manuscript, they write: "We found that none of these features, either alone or in linear combination, correlated with overall transcriptional output" (page 8). However, the Pearson correlation coefficient for the multiple linear regression shown in Fig. S2 ranges from 0.4-0.55 at higher E2 concentrations. In other parts of the manuscript, for example when comparing total RNA in the first and second half of an experiment, a similar range of PCCs (0.47 - 0.6) is called "highly correlated" (response to my point 2.d). Maybe the authors can balance their definition of correlation a bit or provide a more detailed explanation.

2nd Revision - authors' response

16 January 2018

Reviewer #1:

The authors have done a good job responding to my comments, and the manuscript looks great. Arjun Raj

Reviewer #4:

In general, the authors have very well addressed my comments to the initial manuscript, specifically by adding independent validation through smFISH.

In their response to point 2.c (reviewer 3 comment 2), they mention that "sister cells have a higher correlation of total RNA output in the first 6h after mitosis compared to randomly selected cells". I may have missed something, but as far as I can see, only the correlation for sister cells is shown in the corresponding figure S3B. If this is true, maybe the authors can add the data for randomly selected cells as comparison.

We added points of randomly paired sister cells to the scatter plot in Figure S3B, and provide the corresponding correlation coefficient.

In response to the same issue, the authors analyze the correlation of transcriptional outcome to combinations of morphological features and claim that they "did not find very strong predictive power". In the manuscript, they write: "We found that none of these features, either alone or in linear combination, correlated with overall transcriptional output" (page 8). However, the Pearson correlation coefficient for the multiple linear regression shown in Fig. S2 ranges from 0.4-0.55 at higher E2 concentrations. In other parts of the manuscript, for example when comparing total RNA in the first and second half of an experiment, a similar range of PCCs (0.47 - 0.6) is called "highly correlated" (response to my point 2.d). Maybe the authors can balance their definition of correlation a bit or provide a more detailed explanation.

We agree with the reviewer that our interpretation of the correlation coefficient was not entirely consistent. We therefore toned down our conclusion concerning the impact of morphological features on transcriptional outcome in the main text (p.8, second paragraph), and discuss only the weak correlation of individual features in the revised manuscript.

We decided to remove the stronger correlation of a linear combination of features with transcriptional output from the main text, mainly because the features contributing to this linear combination were not robust across estrogen doses (Figure S2). We nevertheless discuss the regression results in the revised caption of Figure S2 and state that "the Pearson correlation coefficient for the multilinear regression is higher (0.4-0.55 at higher E2 concentrations) than for individual features". Moreover, we mention that features contributing to the regression vary across estrogen doses.

In conclusion, we no longer term the Pearson correlation coefficient for the multiple linear regression weak, thereby addressing the reviewers' comment.

Corresponding Author Name: Stefan Legewie

Manuscript Number: MSB-17-7678